# DISCOVERY OF DIVERSE AND REALISTIC FINANCIAL TAIL-RISK USING GENERATIVE FLOW NETWORKS

## ABSTRACT

Accurate modeling of rare high-impact financial events is essential for robust risk management, but existing methods face notable challenges. Sampling-based techniques like Sequential Markov Chain Monte Carlo generate diverse scenarios but often lack focus on specific high-risk outcomes, leading to inefficient exploration. In contrast, Deep Reinforcement Learning methods optimize sequential decision-making effectively but tend to converge on a narrow set of high-reward scenarios, limiting diversity. To address these shortcomings, we propose using Generative Flow Networks (GFlowNets), which naturally learn to sample according to a predefined reward distribution. Our approach systematically generates diverse and realistic tail-risk scenarios by explicitly defining rewards that prioritize high-risk, impactful financial scenarios. Experiments on real-world financial data show that our GFlowNet-based method significantly enhances scenario diversity and realism, effectively capturing the complex nonlinear dependencies typical of financial markets. These findings demonstrate the potential of GFlowNets as a robust framework for financial risk modeling, offering actionable insights into rare but critical market events.

## 1 INTRODUCTION

Effective financial risk management hinges upon anticipating tail risks—rare, extreme events that reside in the far ends of a probability distribution and result in disproportionately large financial losses. These tail events, such as market crashes, systemic failures, or liquidity crises, are characterized by their low frequency yet high impact, posing substantial challenges to both prediction and mitigation. Traditional risk assessment techniques, including historical simulation and Value-at-Risk (VaR), often fall short in modeling such events. These methods rely heavily on historical data and typically assume Gaussian or otherwise well-behaved distributions, thereby underestimating the likelihood and severity of extreme outcomes. Consequently, they provide limited visibility into tail risk and leave financial institutions vulnerable to severe, unforeseen shocks.

Recent advancements have sought to overcome these limitations. Sampling-based methods, such as Sequential Markov Chain Monte Carlo (SMCMC), naturally produce diverse scenarios but lack the ability to efficiently target high-risk regions of the state space, resulting in low sample efficiency. Deep Reinforcement Learning (DRL) approaches, while adept at discovering high-reward sequential strategies, tend to converge to narrow sets of optimal behaviors, often sacrificing the diversity necessary for robust scenario analysis.

This paper introduces Generative Flow Networks (GFlowNets) as a compelling alternative for tail-risk scenario discovery. GFlowNets learn a stochastic policy that samples full trajectories proportionately to a reward function, enabling principled exploration of high-impact outcomes. By designing reward functions that prioritize rare and adverse financial states, our framework generates diverse, realistic, and high-severity scenarios that existing methods often fail to capture. This ability makes GFlowNets a powerful tool for stress testing, crisis prediction, and systemic risk modeling in financial systems.

The broader impact of utilizing GFlowNets for discovering financial risk scenarios extends beyond finance. Their capacity to efficiently explore complex, high-dimensional functions makes them highly applicable in domains such as drug discovery, materials science, and system optimization, where diversity and coverage of solutions are crucial. By enabling the unsupervised and diverse

generation of effective scenarios—especially in spaces with continuous and interdependent parameters—GFlowNets can drive significant advances in robust design and anomaly detection. Since many real-world processes are governed by functions that are intractable, multi-modal, or poorly understood, the ability to systematically uncover critical scenarios or solutions enhances not only the reliability of financial risk management but also offers pathways to transformative breakthroughs in science and engineering. Our primary contributions can thus be summarized as follows:

- We introduce a GFlowNet-based approach specifically designed to address the challenge of generating diverse and realistic financial risk scenarios by leveraging proportional sampling relative to a reward distribution.
- Our approach is the first to successfully apply GFlowNets to large continuous state spaces, enabling the use of the architecture to previously unexplored problems. Our experiments show that our approach is highly effective for exploring complex continuous functions.
- We empirically validate our proposed framework on real-world financial data, demonstrating superior performance in terms of scenario diversity, realism, and capturing nonlinear financial dependencies.

## 2 RELATED WORK

We review key methodologies relevant to financial risk scenario modeling, emphasizing their ability to discover diverse, high-risk trajectories. Existing approaches can be broadly grouped into three families: deep generative models, reinforcement learning techniques, and probabilistic simulation.

**Deep Generative Models** such as GANs Zhang et al. (2018), VAEs Desai et al. (2021), diffusion models Yuan & Qiao (2024) and LLMs Cao et al. (2024); Yuksel & Sawaf (2025) are increasingly used in finance for tasks such as synthetic market data generation and anomaly detection. While GANs are effective at generating realistic sequences, they often suffer from mode collapse and are not designed to explicitly model rare or tail-risk scenarios, limiting their applicability in high-diversity risk discovery.

**Reinforcement Learning (RL)** has been widely applied to financial modeling, portfolio optimization Cui et al. (2024), and risk management Anwar & Zhang (2024). However, traditional RL methods, including DQN, policy gradients, and actor-critic algorithms, prioritize the optimization of expected returns. This leads agents to converge on a small set of optimal strategies, hindering their ability to explore diverse failure modes. While DRL is not commonly used for risk discovery, it is frequently applied in adjacent tasks. Nonetheless, the tendency of neural agents to collapse to narrow behavioral policies makes them less suited for sampling rare events. We empirically validate these limitations in Section 5.

**Probabilistic Simulation Methods** such as Sequential MCMC (SMCMC) Andrieu et al. (1999), jump diffusion models Runggaldier (2003), and Bayesian Optimization (BO) Shahriari et al. (2015) offer more interpretable and uncertainty-aware alternatives. SMCMC extends classical MCMC with temporal dynamics, producing diverse scenario trajectories, but incurs high computational cost and slow convergence in high-dimensional spaces. Jump diffusion models embed rare events into stochastic processes for stress testing, yet require strong parametric assumptions and lack data-driven adaptability. BO has shown promise in identifying extreme outcomes with minimal evaluations by optimizing risk-aware acquisition functions Frazier (2018); Cheon et al. (2024), but it typically focuses on global optima, limiting diversity in discovered trajectories.

## 3 BACKGROUND

GFlowNets offer a framework for learning a generalized stochastic policy Atanackovic & Bengio (2024) that constructs complex objects through sequential actions, with the goal of sampling each object $x$ with probability proportional to a given non-negative reward $R(x)$ rather than maximizing cumulative reward as in standard reinforcement learning Bengio et al. (2021). By framing generation as a flow network over a directed acyclic graph of partial states, GFlowNets convert an unnormalized target distribution into an amortized sampler that produces independent, diverse samples in one shot, avoiding the slow mixing and mode-collapse issues of MCMC and the limited diversity of deep RL methods Bengio et al. (2023).

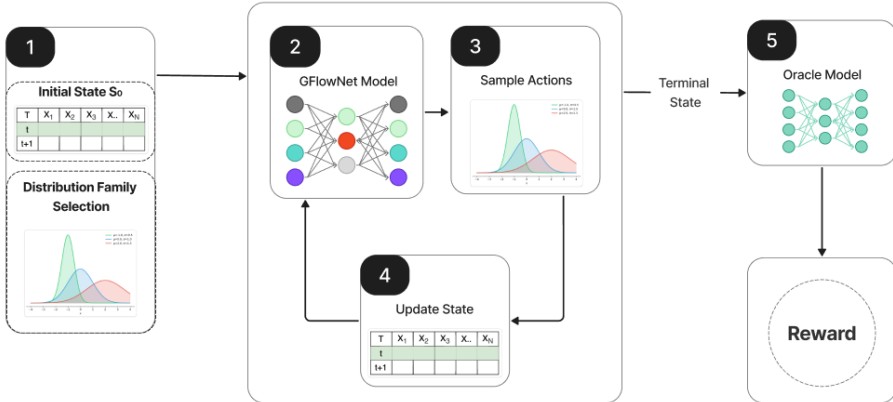

Figure 1: An overview of our proposed method. GRID begins by modeling each environmental variable and the initial state, then explores the state space by sampling trajectories. The sampling is guided using GFlowNet's loss function and feedback from our Oracle.

At the core of a GFlowNet is a flow function $F$ defined over edges $(s \rightarrow s')$ in the state graph, which must satisfy a conservation (consistency) constraint at every intermediate state

$$s = \sum_{s': (s,s') \in E} F(s \rightarrow s') = \sum_{s'': (s'',s) \in E} F(s'' \rightarrow s) \tag{1}$$

ensuring that the total inflow equals the total outflow except at the initial and terminal states Bengio et al. (2021). The flows to terminal states are set equal to their rewards, so that the total flow through a terminal state coincides with $R(x)$. From these edge flows, a forward policy is derived as

$$\pi(s' \mid s) = \frac{F(s \rightarrow s')}{F(s)}, \quad F(s) = \sum_{s':(s,s') \in E} F(s \rightarrow s'), \tag{2}$$

which guarantees that the induced marginal probability of generating any terminal state $x$ is

$$P_F(x) = \frac{R(x)}{\sum_{x'} R(x')}, \tag{3}$$

This alignment between the learned policy and the reward distribution enables GFlowNets to sample high-reward scenarios more frequently while still maintaining support over all modes of the target Bengio et al. (2023).

Three properties make GFlowNets particularly effective for modeling financial risk scenarios:

- **Amortized one-shot sampling:** Once trained, the GFlowNet enables efficient generation of diverse and independent scenarios in a single forward pass, eliminating the need for costly iterative procedures like MCMC or sequential simulation, and making it well-suited for large-scale stress testing.
- **Reward-aligned exploration:** The sampling distribution is proportional to a predefined reward function, directing the generator toward high-impact or extreme-risk scenarios. This reward-driven inductive bias improves tail event coverage compared to uniform Monte Carlo and mitigates the mode collapse often seen in RL-based generative models.
- **Multi-modal distribution modeling:** GFlowNets naturally support exploration of complex, multi-peaked distributions. This is critical for financial risk where multiple distinct failure modes (e.g., liquidity crises vs. inflation shocks) may exist and must be captured simultaneously without bias toward a single dominant mode.

Recent advances extend GFlowNets to continuous, high-dimensional domains Lahlou et al. (2023); Brunswic et al. (2024) using support measures and transition kernels. Additional training objectives such as Detailed Balance, Trajectory Balance, and Sub-Trajectory Balance Malkin et al. (2023);

---

**Algorithm 1** Training GFlowNet for Risk Scenario Generation

---

1: **Input:** Initial state $s_0$, oracle $\mathcal{O}$, iterations $N$, buffer size $\mathcal{C}$, trajectory length $T$, distribution family $\mathcal{D}_{\text{family}}$
2: **Initialize:** Forward policy $\pi_F$, backward policy $\pi_B$, replay buffer $\mathcal{D}$
3: **for** $i = 1$ to $N$ **do**
4:     Reset to $s_0$, initialize empty trajectory
5:     **for** $t = 1$ to $T$ **do**
6:         Sample action $a_t \sim \pi_F(s_t)$
7:         Update state $s_{t+1} = f(s_t, a_t)$
8:         Store $(s_t, a_t, s_{t+1})$ in trajectory
9:     Predict outcome: $\hat{y} = \mathcal{O}(s_T)$
10:    Compute reward: $R = \max(0, -\hat{y})$
11:    Add trajectory $(s_0, \ldots, s_T, R)$ to buffer $\mathcal{D}$
12:    **if** $|\mathcal{D}| > \mathcal{C}$ **then**
13:        Remove oldest trajectory
14:    Sample batch of trajectories from $\mathcal{D}$
15:    **for** each trajectory **do**
16:        Compute trajectory balance loss $L_{\text{TB}}$
17:        $L_{\text{TB}} = (\log Z + \sum \log \pi_F - \sum \log \pi_B - \log R)^2$
18:        Update $\pi_F, \pi_B$ via gradient descent on $L_{\text{TB}}$
19: **Return:** Trained policy $\pi_F$

---

Madan et al. (2023), which enforce flow consistency across different granularities, also contribute to the architecture's performance.

# 4 RISK SCENARIO GENERATION USING GFLOWNETS

**Overview.** We present our **GF**lowNet-based **RI**sk **D**etection (GRID) method in Figure 1. We begin by specifying the initial macroeconomic state, a parametric distribution family for each environmental variable (to be learned by the network), and a fixed trajectory length to define termination (step #1). At each step, the GFlowNet predicts parameters governing the action distribution (step #2), samples a transition (step #3), and updates the environment state by applying the sampled change to the current state (step #4). This process is repeated iteratively to construct a full trajectory. Upon reaching the terminal state, a predictive oracle evaluates the trajectory and provides a scalar reward used to train the generative policy via flow-matching objectives (step #5).

The proposed architecture has three important advantages over existing scenario generation methods. First, it provides strong explorative capabilities, particularly when using flexible distribution families such as mixtures of Gaussian or Beta functions. Secondly, GRID seamlessly supports the use of different distribution functions for each environmental variable. This flexibility enables our approach to accurately model the environment and facilitates exploration of edge-cases. Standard energy- or diffusion-based algorithms do not support this capability, making its implementation challenging. Third, once trained, the policy acts as an amortized sampler, enabling rapid generation of high-reward scenarios without iterative inference, unlike conventional SMCMC or optimization-based approaches.

## 4.1 INITIALIZING STATE AND ENVIRONMENT VARIABLE DISTRIBUTIONS

**The initial state.** The initial state of our process begins at a specified macroeconomic state, comprising of core financial indicators such as equity indices and trading volumes and a timestep $t$. A full list of the variables of our environment is provided in Table 1. This set of variables enables our model to create an accurate simulation of real-world scenarios.

**Modeling the environment's variable distribution.** To take advantage of GRID's ability to utilize multiple distribution types at the same time, we define a set of possible distributions to model each environmental variable. In our implementation, we support the following distribution functions: *Gaussian*, *Gaussian Mixture*, *Beta*, *Mixture Beta*, and *Student-t*.

Our selection of possible distributions is guided by the variables of our environment. For example, the VIX closing value, bounded and asymmetric, might require a Beta distribution, whereas long-term US interest rates are well-modeled by Gaussian mixtures around a stable mean. By combining different distribution functions, we ensure that our generated scenarios can adapt to observed skewness, bounds, and tail risk. Once the distribution type has been defined for each variable, the hyperparameters of that distribution are modified as part of the network's learning process. This means that in cases

Table 1: Initial macroeconomic environment parameters used to define the starting state $s_0$.

| Variable Name | Description |
| --- | --- |
| Volume_spx | Trading volume of the S&P 500 index |
| Close_ndx | Closing price of the NASDAQ-100 index |
| Volume_ndx | Trading volume of the NASDAQ-100 index |
| Close_vix | Closing value of the VIX (volatility index) |
| IRLTCT01USM156N | Long-term US interest rate (inflation/deflation proxy) |
| BAMLH0A3HYCEY | High-yield corporate bond spread (credit risk indicator) |

where extreme values – specifically, those that induce greater risk – yield higher rewards, the eventual distribution will give those edge cases a somewhat greater probability of being explored.

When using a mixture model (Gaussian or Beta) to model a variable, GRID adjusts not only the parameters of each of the model's $K$ individual distributions, but also the set of weights responsible for the latter's weighting and integration. For each action variable, the policy predicts raw logits $\mathbf{w} \in \mathbb{R}^K$, means $\boldsymbol{\mu} \in \mathbb{R}^K$, and standard deviations $\boldsymbol{\sigma} \in \mathbb{R}^K$. Mixture weights are computed via a temperature-controlled Gumbel-Softmax Jang et al. (2016):

$$\boldsymbol{\alpha} = \text{GumbelSoftmax}(\mathbf{w}, \tau), \tag{4}$$

where $\tau$ is a learnable or fixed temperature. The final action distribution for each variable is a weighted sum over component parameters:

$$a \sim \mathcal{N}\left( \sum_{k=1}^{K} \alpha_k \mu_k, \ \sum_{k=1}^{K} \alpha_k \sigma_k \right), \tag{5}$$

producing a unimodal approximation that retains multimodal expressivity during training. This dynamic mixture formulation improves trajectory diversity and supports feature-specific heteroscedasticity and temporal variation in action uncertainty.

## 4.2 Implementing State Transitions

At each timestep, the sampled action – parameterized as a distribution over economic variable changes – is applied to the current macro-financial state to produce the next. These actions represent the changes to each of our environmental variables (credit spreads, interest rates, equity indices, etc). Formally, given state $s_t$ and action $a_t$, we define the next state as $s_{t+1} = f(s_t, a_t)$ under affine transformation and domain clipping constraints.

To ensure that each state transition remains meaningful over time, and to prevent repetitive cyclic trajectories, we incorporate a *time-index* into the state representation. This addition guarantees that even if the same state occurs twice, the trajectory does not degenerate into loops because $(s_t, t) \neq (s_{t'}, t')$ for $t \neq t'$. This design encourages diverse sequential paths and enables the model to learn time-dependent dynamics in the action distributions (for further details, see Figure 6 in the supplementary material).

Another important aspect of our model is ensuring that state transitions are plausible based on historical data. Simply put, while we need to explore diverse scenarios, we also need to ensure that changes in environmental variables are not unreasonable (e.g., a change of 20% in interest rate in a month). To achieve this goal, we randomly sampled a large number of sequential time steps from our (real-world) training set, and identified the maximal mean and standard deviation observed in a *single step*. When using GFlowNet to generate our trajectories, we limit our model by not allowing it to generate state transitions with greater fluctuations than those observed by our sampling. It is important to note that the mean and standard deviation we use when limiting our model are not necessarily derived from the same sequence, meaning that allow *greater flexibility* than what we observed in historical data, but not without limitations.

## 4.3 Terminal State Evaluation and Reward

Once a trajectory reaches its terminal state $s_T$, it is evaluated by a pre-trained oracle model that predicts a downstream financial outcome. In our case, the outcome is the projected 12-month

percentage return of the S&P 500 index. The output of this oracle, denoted $\mathcal{O}(s_T)$, is then mapped to a scalar reward via:

$$R(s_T) = \max(0, -\mathcal{O}(s_T)) \tag{6}$$

This reward formulation assigns zero value to positive market movements while positively rewarding trajectories that lead to negative returns—thus aligning with the objective of discovering plausible but adverse financial evolutions. During training, we used a dense neural network as our Oracle, and during training we used an ensemble of learning models. Although all models were trained on the same training data, the use of different models prevents all risk models from benefiting from overfitting. A detailed description of the Oracles' architecture and training is presented in Section 5.2.

### 4.4 Training Objective and Replay Buffer.

The GFlowNet is optimized using the *trajectory balance loss*, which enforces that the likelihood of sampling a trajectory is proportional to its final reward. The loss is defined as:

$$\mathcal{L}_{\text{TB}} = \left( \log Z + \sum_{t=0}^{T-1} \log \pi_F(s_{t+1} \mid s_t) - \sum_{t=1}^{T} \log \pi_B(s_{t-1} \mid s_t) - \log R(s_T) \right)^2, \tag{7}$$

where $\pi_F$ and $\pi_B$ denote the forward and backward policies, respectively, $Z$ is the partition function (a normalizing constant that ensures probabilities over all possible outcomes sum to one), and $R(s_T)$ is the reward assigned to the final state of the trajectory. To enhance sample efficiency and training stability, we incorporate a replay buffer Mnih et al. (2013) that stores past trajectories and allows for resampling, thereby smoothing gradient estimates and mitigating variance.

## 5 Evaluation & Analysis

### 5.1 Datasets and Baselines

**Datasets.** We constructed a macro-financial dataset using publicly available sources (FRED[1], Yahoo Finance) covering the years 1996-2023. We extracted the features presented in Table 1 in monthly intervals. All features were normalized to a $[-100, 100]$ scale. The 27-year time covered by our dataset contains three major financial crises: *a)* the Dot-com bubble (2001–2002), *b)* the Global Financial Crisis (2007–2008), *c)* and the COVID-19 stock market crash (2020–2022). We use the time-span of these crises as our test sets.

**Baselines.** We compare GRID to three established scenario discovery baselines: SMCMC, REINFORCE, and Soft Actor-Critic. As we explain in Section 2, these baselines cover the main approaches in risk scenario exploration. *All methods operate under the same trajectory horizon, oracle reward function, and action space*.

**1) GRID** – Our GFlowNet was trained using the trajectory balance loss, with a fixed reward function defined as $R(s_T) = \max(0, -\mathcal{O}(s_T))$. Forward and backward policies were parameterized as 3-layer MLPs with LeakyReLU activations and layer normalization. Trajectories consisted of 12 timesteps, and training was stabilized using a trajectory replay buffer with a size of 5000. We used the Adam optimizer with a learning rate of $5 \times 10^{-4}$ for both forward and backward networks, as well as for the learnable partition function $\log Z$. We used a batch size of 16 in all experiments. After training, we sampled 200 trajectories for evaluation.

**2) Sequential MCMC (SMCMC)** – SMCMC samples trajectories using a Gaussian proposal with standard deviation 10 and Boltzmann temperature 10. For each evaluated use case, we generated 2,000 trajectories and retained the top 200, ranked by oracle-predicted reward, for evaluation. This method is fully offline and does not involve learning dynamics or gradient-based optimization.

**3) REINFORCE** – this baseline uses a 3-layer MLP actor (`hidden dim = 128`, LeakyReLU activations) to predict the parameters (mean and standard deviation) of a Gaussian policy. Policies are trained using the REINFORCE objective with trajectory-level rewards. The algorithm was trained for 1,500 episodes After training, the model generated 200 trajectories for evaluation.

---

[1]https://fred.stlouisfed.org/

Table 2: Oracle performance (MLP) on Dot-Com (2002), GFC (2008), and COVID-19 (2021). The metrics are evaluated over a 24-month validation period. Lower MAE/RMSE is better; higher $R^2$ is better.

| Metric | Dot-Com (2002) | GFC (2008) | COVID-19 (2021) |
|---|---|---|---|
| $R^2 \uparrow$ | -0.4762 | 0.0942 | -0.8638 |
| RMSE $\downarrow$ | 11.59 | 4.2048 | 14.3532 |
| MAE $\downarrow$ | 9.78 | 2.92 | 12.81 |

**4) Soft Actor-Critic (SAC)** – We used a Gaussian policy and twin Q-networks. The policy is modeled by a 2-layer MLP (`dim = 64`), and each critic is a 2-layer MLP (`dim = 128`). Target Q-networks are updated with $\tau = 0.0005$, and the entropy temperature was automatically tuned from an initial value $\alpha = 0.5$. We used a discount factor $\gamma = 0.999$. SAC was trained for 1,500 episodes, ensuring convergence. After training, 200 trajectories were sampled from the learned policy for evaluation.

It is important to note that we experimented with multiple hyperparameter configurations and training times for the baselines. *For each model, we report the top-performing configuration*. Furthermore, additional training did not improve the baselines' performance.

## 5.2 EXPERIMENTAL SETUP

**Oracle Pretraining.** We trained our predictive oracle model on historical macro-financial data to estimate future market returns. Training was conducted on an RTX 3070 Ti GPU and an AMD Ryzen 9 7950X (16-core) CPU. The oracle's training set features were jointly defined with the company on whose model we were relying, in order to ensure that our oracle focuses on risk-associated aspects. The oracle's outputs define the scalar rewards used by the GFlowNet. To ensure strict temporal integrity and eliminate any risk of data leakage, the oracle was trained exclusively on pre-crisis data for each evaluation scenario. For example, in the Dot-com bubble case, the model was trained on data from 1996 to 2000 and used only for evaluating trajectories beginning in 2001. Each trajectory spans a 12-month horizon, with monthly transitions capturing directional shifts in economic variables. The performance metrics of our Oracle are presented in Table 2. This setup guarantees that the GFlowNet explores risk scenarios strictly forward in time from the oracle's training window.

**Evaluation Metrics.** We used two metrics to measure he financial losses associated with the discovered scenarios: (i) *Median reward* – the median of oracle-predicted 12-month S&P-500 change (higher is riskier). The median provides a more accurate evaluation of performance than the average, due to potential extreme values; (ii) *Q90 reward* – the reward of the trajectories of the 90th quintile.

**Oracle models.** We intentionally train with a single MLP Oracle and evaluate using an ensemble Oracle (Random Forest, Gradient Boosting, Elastic Linear Model, MLP) to ensure the GFlowNet does not overfit to one model's inductive biases. This enforces robust stress discovery and reflects practical finance settings, where risk estimates are inherently model-dependent and best practice involves conservative aggregation via ensemble methods. If generated scenarios yield high risk under the ensemble, tail discovery is more credible and transferable.

This design also serves as an implicit ablation: if the GFlowNet learned only the training Oracle's boundaries, performance would deteriorate under the ensemble. As shown in Tables 3 and 4, our approach maintains strong discovery of tail events under both MLP and ensemble test Oracles, validating the robustness of the method. For completeness, full metrics for the ensemble Oracle ($R^2$, RMSE, MAE) are provided in the appendix.

**Training Phase Oracle Architecture.** The oracle model is a lightweight 3-layer MLP with 32 hidden units per layer, LeakyReLU activations, and layer normalization. It was trained using the Adam optimizer (learning rate $10^{-4}$) for 300 epochs to minimize mean squared error (MSE).

**Distribution Selection.** We experimented with multiple action distribution families, including Gaussian, Student's t, and Beta. To allow flexible bounded transitions, Beta distributions were transformed via affine mappings into the $[-30, 30]$ range. In practice, we found that mixture distributions significantly improved diversity and expressiveness.

Table 3: Performance metrics (Median and Q90) over 30 independent runs per algorithm on three financial crises (2002, 2008, 2021). Higher is better. Rewards are clipped at 100.

| Method | Median Reward | | | Q90 Reward | | |
|---|---|---|---|---|---|---|
| | 2002 | 2008 | 2021 | 2002 | 2008 | 2021 |
| GFlowNet | **43.11** | **31.13** | 18.12 | 58.57 | 54.52 | **100.00** |
| REINFORCE | 32.02 | 19.37 | 24.33 | **65.57** | 57.06 | 98.53 |
| SAC | 5.34 | 18.86 | 34.00 | 40.99 | **67.92** | **100.00** |
| SMCMC | 26.01 | 29.62 | **42.71** | 30.52 | 34.22 | 49.03 |

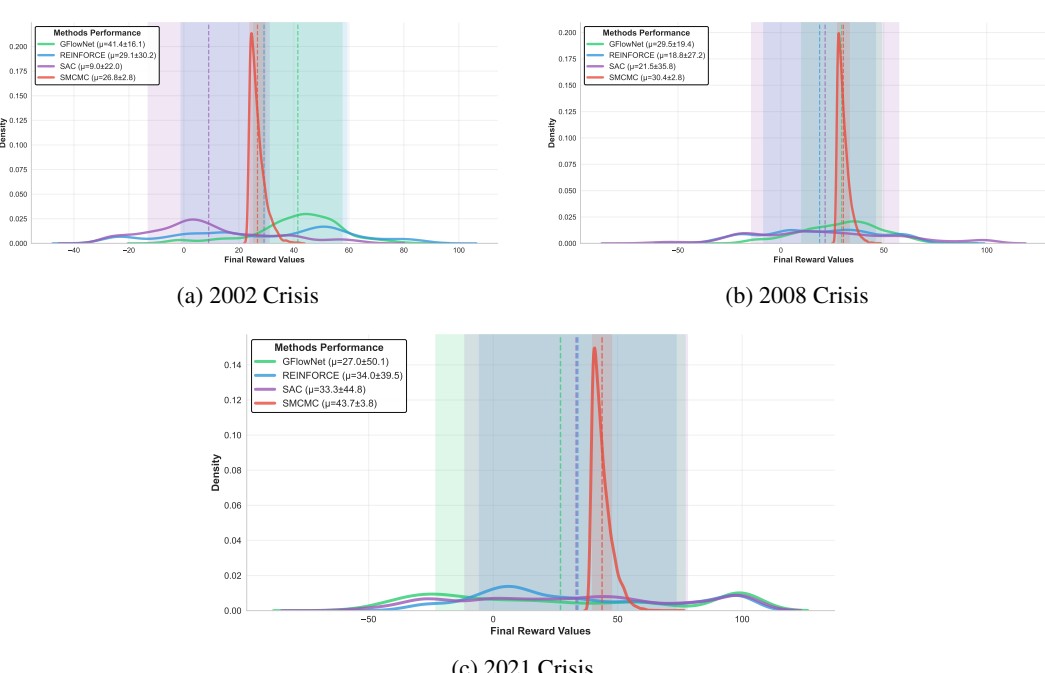

(a) 2002 Crisis

(b) 2008 Crisis

(c) 2021 Crisis

Figure 2: Distribution of scenario rewards generated by each algorithm across 30 independent runs (total of 6,000 trajectories per method) for three financial crises. Each ridge represents the density of final rewards. Overlaid lines show the mean and shaded area represents the mean +- 1 std.

## 5.3 EVALUATION RESULTS

We performed 30 independent runs per algorithm, each initialized with a different random seed. For each run, we trained a new model (except for SMCMC, which is non-trainable), and sampled 200 trajectories using the final learned policy. For SMCMC, we generated 2000 trajectories per run and selected the top 200 according to the oracle's reward, for fair comparison with the learned approaches.

The results in Table 3 demonstrate that GRID consistently outperforms competing methods overall. Our approach achieves the highest median rewards in two out of the three evaluated crises. In the 2021 case, although the median reward is lower than competitors, the reward distribution is bimodal, which depresses the median. Notably, GRID still achieves the highest performance at the 90th percentile, indicating strong tail performance.

In the Q90 Rewards metric, GRID achieved top performance for the 2021 use-case (tied with SAC), a fact that supports the analysis presented above. While our approach did not achieve top results in this metric for the 2002 and 2008 use-cases, the rewards are high and indicate GRID's ability to generate multiple high-reward scenarios. This stands in sharp contrast to SAC and REINFORCE, which achieved top scores in the Q90 metric, but whose median rewards for the same use-cases are considerably lower than GRID's. These results indicate that while the DRL-based baselines are effective in discovering high-reward trajectories, they are not able to discover many of them.

Table 4: Diversity metrics across three financial crashes. We present raw and normalized diversity scores for full trajectories (DTW-based) and terminal states. Higher is better. Results are from 100 trajectories per method.

| Method | Traj Div (DTW) | | | Norm. Traj Div | | | Terminal-State Div | | | Norm. Terminal-State Div | | |
|---|---|---|---|---|---|---|---|---|---|---|---|---|
| | 2002 | 2008 | 2021 | 2002 | 2008 | 2021 | 2002 | 2008 | 2021 | 2002 | 2008 | 2021 |
| GFlowNet | 108.85 | **550.39** | **399.15** | 99.02 | **481.41** | **399.15** | 79.80 | 266.13 | **200.81** | 72.60 | 232.78 | **200.81** |
| REINFORCE | 94.31 | 402.24 | 229.17 | 90.71 | 352.80 | 229.17 | 77.24 | **271.92** | 160.71 | 74.29 | **238.50** | 160.71 |
| SAC | **260.25** | 136.10 | 264.38 | **244.84** | 136.10 | 264.38 | **182.53** | 92.27 | 187.60 | **171.72** | 92.27 | 187.60 |
| SMCMC | 182.01 | 186.48 | 164.81 | 163.69 | 167.92 | 139.29 | 100.93 | 103.47 | 96.53 | 90.77 | 93.17 | 81.58 |

Finally, we plot the scenario distribution of all the baselines in Figure 2. The graphs clearly show that GRID offers the best trade-offs between diversity and high rewards. While SMCMC discovers multiple diverse scenarios with reasonable rewards, it has difficulty discovering trajectories with very high risk. The DRL-based baselines are able to detect these scenarios, but at the expense of diversity.

## 5.4 ANALYZING RISK SCENARIO DIVERSITY

Our goal is to determine whether GRID's high performance comes at the cost of trajectory diversity. We now analyze the *diversity of the generated scenarios*, both with respect to the terminal state $S_T$, and the trajectories used to achieve them. We define two metrics:

**Terminal-State Diversity -** We calculate the mean of the Euclidean distances between every pair of terminal states. This metric enables us to evaluate each model's diversity of the outcomes.

$$\text{Diversity}_{\text{Last}} = \frac{2}{N(N-1)} \sum_{i=1}^{N} \sum_{j=i+1}^{N} \|s_T^{(i)} - s_T^{(j)}\|_2,$$ 

(8)

**Trajectory Diversity -** We calculate the mean of the Dynamic Time Warping (DTW) Sakoe & Chiba (2003) metric for the trajectories of every pair of discovered scenarios. DTW is commonly used to evaluate the similarity in time series data, and we use it to evaluate the diversity of the path instead of only the outcome:

$$\text{NormalizedScore} = 1 - \frac{|r_{\max} - r_{\min}|}{|r_{\max}| + |r_{\min}| + \epsilon}$$ 

(9)

where , $r_{\max}$ and $r_{\min}$ are the maximal and minimal rewards obtained by each model's scenarios. This metric complements the one above, and together they provide a holistic assessment of each model's diversity. We also calculate *normalized versions* of our two metrics, which factors reward stability, measuring the model's ability to consistently discover high-reward scenarios. See Table 4 for results.

GRID performs well in terms of trajectory diversity, achieving the top results for the 2008 and 2021 use cases (both normalized and not). In the terminal state diversity, our approach achieves top performance for the 2021 use-case, while remaining highly competitive in the other two use-cases. It is important to remember that SAC and REINFORCE achieve high diversity mostly due to their focusing on a few high-quality trajectories, and generally achieving lower performance in all others. In conclusion, this analysis shows that GRID is able to maintain high levels of trajectory diversity while generating trajectory populations that outperform those generated by the baselines.

## 6 CONCLUSION, LIMITATIONS, AND FUTURE WORK

We present GRID, a GFlowNet-based approach for discovering financial tail-risk scenarios. It balances exploration and exploitation, identifying diverse high-risk cases beyond state-of-the-art baselines. Our key contribution is the first application of GFlowNets to large continuous state–action spaces, enabled by techniques such as Gumbel Softmax and specialized sampling.

GRID has limitations: its effectiveness depends on oracle quality, and exploration is guided implicitly by the learned policy without direct control of exploration–exploitation or computational constraints. Nonetheless, exploration can be steered through design choices, such as output distribution families, mixture model components, and bounded parameters. Future work will address these limitations and extend applications to neural architecture search, hyperparameter optimization, and feature selection.

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

## A  LLM USAGE

LLM was used for copy-editing to improve grammar and phrasing, but not for generating scientific content or code.

## B  SUPPLEMENTARY RESULTS

### B.1  VISUALIZING TRAJECTORY AND TERMINAL STATE DIVERSITY

We now expand on our analysis in Section 5.4 by presenting visualizations of the diversity of the trajectories and terminal states generated by our top-performing methods.

In Figure 3, we compare the terminal state diversity of GRID to those of SAC and SMCMC. Distances were calculated using Euclidean distance, as described in Section 5.4. Our results show that our approach and SMCMC generate much more diverse terminal states compared to SAC. These results further support our conclusion that DRL-based methods are effective for optimization but not diversification.

In Figure 4, we present the pairwise DTW distances for top-performing trajectories. Once again, the results show that our GFlowNet-based approach consistently discovers a broader spectrum of plausible economic evolutions, balancing both exploration and high-reward fidelity (as shown in our results).

## C  MODELING FEATURE DISTRIBUTION DIVERSITY

**Distribution diversity for environmental variables.** In Figure 5, we present the final distributions of our environment's variables, learned by our GFlowNet-based approach. It is clear that each distribution is different from the others, demonstrating GRID's ability to create feature-specific modeling.

**Distribution diversity for a single variable over time.** In Figure 6 we present the distribution of a single variable in our environment, *modeled for each time step* in our 12-step trajectories. As shown by the figure, the distribution changes significantly over time, as our model calibrates its distributions to produce the most effective (i.e., high-risk) scenarios.

## D  FULL REWARD DISTRIBUTIONS FOR MLP ORACLE

We report here the full distributions of final rewards for each method across all three use-cases (2002, 2008, 2021) when trained with the MLP oracle. These plots complement the summary statistics in Table 3 by illustrating the entire empirical distribution of rewards across 6,000 trajectories per method.

Visual inspection confirms that GRID tends to produce a large mass of high-reward trajectories, though in some cases (notably 2021) the reward distribution is multimodal. This explains why the median reward is lower relative to competitors despite superior tail performance, as captured by the 90th percentile results. The plots for the 2002, 2008, and 2021 use-cases are presented in Figures 7, 8, and 9 respectively.

### D.1  2008 USE-CASE

### D.2  2021 USE-CASE

## E  TRAJECTORY SIMILARITY TO ACTUAL EVENTS

To test whether our approach can generate realistic scenarios, we analyzed all methods' capabilities of predicting the actual sequence of events for each of three financial crises presented in our study (2002, 2008, 2021). For each scenario, we used Person correlation to calculate the trajectory similarity of the top-10 and top-50 rated (risk score-wise) trajectories to the events that happened in the real world.

Note that correlation was calculated over the entire trajectory, and not only the final outcome ( "how we got here?" is more important than "where are we?").

The results of our analysis are presented in tables 5-7. We present the maximal and mean similarities, as well as the standard deviation. The results show that our approach achieves the top similarity in at least one of the two evaluated cases for each scenario. Moreover, the standard deviation of our trajectories' similarity is larger than those of all methods except MSMCM (which is logical, since this baseline fully focuses on diversity). High diversity is a positive. meaning our approach discovers a larger array of risks.

| Method | Top_K | Max_Sim | Mean_Sim | Std |
|---|---|---|---|---|
| GFlowNet | 10 | **0.374** | **0.267** | 0.073 |
| | 50 | 0.380 | **0.265** | **0.084** |
| REINFORCE | 10 | 0.108 | 0.056 | 0.078 |
| | 50 | 0.136 | 0.066 | 0.054 |
| SAC | 10 | -0.179 | -0.243 | 0.044 |
| | 50 | -0.161 | -0.247 | 0.047 |
| SMCMC | 10 | 0.261 | 0.088 | **0.130** |
| | 50 | **0.441** | 0.052 | 0.064 |

Table 5: Trajectory similarity for the 2002 financial crisis. We used Pearson correlation to calculate the similarity of top X-rated (risk score-wise) trajectories with the highest similarity to actual events.

| Method | Top_K | Max_Sim | Mean_Sim | Std |
|---|---|---|---|---|
| GFlowNet | 10 | 0.433 | 0.247 | 0.132 |
| | 50 | **0.562** | 0.276 | 0.133 |
| REINFORCE | 10 | -0.534 | -0.600 | 0.040 |
| | 50 | -0.455 | -0.559 | 0.045 |
| SAC | 10 | 0.416 | **0.374** | 0.024 |
| | 50 | 0.452 | **0.377** | 0.026 |
| SMCMC | 10 | **0.528** | 0.203 | **0.209** |
| | 50 | 0.528 | 0.150 | **0.218** |

Table 6: Trajectory similarity for the 2008 financial crisis. We used Pearson correlation to calculate the similarity of top X-rated (risk score-wise) trajectories with the highest similarity to actual events.

| Method | Top_K | Max_Sim | Mean_Sim | Std |
|---|---|---|---|---|
| GFlowNet | 10 | **0.500** | **0.420** | 0.059 |
| | 50 | **0.533** | **0.430** | 0.067 |
| REINFORCE | 10 | 0.407 | 0.353 | 0.048 |
| | 50 | 0.425 | 0.374 | 0.035 |
| SAC | 10 | -0.124 | -0.143 | 0.013 |
| | 50 | -0.114 | -0.144 | 0.019 |
| SMCMC | 10 | 0.295 | 0.146 | **0.120** |
| | 50 | 0.481 | 0.101 | **0.181** |

Table 7: Trajectory similarity for the 2021 financial crisis. We used Pearson correlation to calculate the similarity of top X-rated (risk score-wise) trajectories with the highest similarity to actual events.

# F    INTERPRETING THE ORACLE $R^2$ PERFORMANCE AND ITS EFFECT ON THE PERFORAMNCE OF OUR APPROACH

## F.1    LOW $R^2$ ARE COMMON IN ASSET PRICING

Our oracle achieves out-of-sample $R^2$ values of -0.48, 0.09, and -0.86 across the three crisis periods (Table 2), which could potentially raise concerns about whether such a predictor can be informative.

However, asset-pricing theory and leading empirical work emphasize that low return $R^2$ is the norm rather than the exception in equity-return prediction, even for models that embed substantial information. In their widely-cited work, Hou et al. (2013) show in a rational-expectations setting that return $R^2$ is essentially independent of how much information is incorporated into prices, so a low $R^2$ does not imply that forecasts are uninformative or that markets are fully efficient.

Empirically, canonical studies on predicting the equity premium—such as Campbell and Thompson (2008) and related work—find statistically significant and economically meaningful predictability with very small out-of-sample $R^2$ statistics. Large-sample studies of cross-sectional and aggregate return predictors Kelly & Pruitt (2013); Stalla-Bourdillon (2022) similarly report $R^2$ values in the same ballpark or below what we observe, while still documenting non-trivial gains in utility or pricing errors. Thus, our oracle's low and occasionally negative $R^2$ values are fully consistent with the broader return-forecasting literature, especially given the added difficulty of forecasting 12-month-ahead crisis-period returns.

By construction, $R^2$ measures unconditional variance explanation and is a coarse proxy for economic usefulness, particularly for long-horizon, regime-dependent outcomes like crisis-period returns. In our setting, the oracle exhibits non-trivial MAE values (2.92–12.81), indicating that it still captures meaningful signal about levels and signs of returns despite explaining little unconditional variance. Leading papers stress exactly this point: small $R^2$ can coexist with sizable economic value, because investors care about tail behavior, state dependence, and utility gains rather than unconditional variance alone Campbell & Thompson (2008); Kelly & Pruitt (2013); Hou et al. (2013).

Related asset-pricing work also documents that low $R^2$ is pervasive and interacts in subtle ways with market efficiency and anomalies Hou et al. (2013); Hu & Liu (2013); Bryzgalova (2015). These results reinforce that low $R^2$ should be viewed as a stylized fact of return data, not as evidence that a model—or, in our case, an oracle used inside a generative framework—is unusable.

### F.2 IMPERFECT ORACLES AND GFLOWNETS

The presence of a noisy, partially misspecified oracle is not a peculiarity of our application but a central design assumption of GFlowNets. GFlowNet Foundations **?** explicitly highlights settings where a cheap proxy for the true reward is available and where diversity of samples is used to hedge against proxy misspecification. In Bengio et al. (2021) the authors introduce GFlowNets precisely to generate diverse high-reward candidates when the "oracle" is uncertain or only loosely correlated with the true objective, and they illustrate that GFlowNets can capture multiple modes even when the true oracle is absent.

Subsequent applications in AI-driven scientific discovery and sequence design Jain et al. (2022); **?** rely on imperfect oracles such as noisy property predictors or approximate physical models, and explicitly argue that diverse samples around the modes of an imperfect oracle increase the chance of finding candidates that satisfy multiple downstream criteria. Our financial tail-risk setting is directly analogous: the return-forecasting oracle is a cheap but noisy proxy for true future outcomes, and the GFlowNet is used to explore the induced reward landscape in a way that is robust to this noise.

### F.3 WHY GFLOWNETS ARE APPROPRIATE WITH A LOW $R^2$ ORACLE

Compared to standard RL algorithms that explicitly maximize expected reward, GFlowNets sample trajectories proportionally to reward, which naturally spreads probability mass over a wide set of high-reward states rather than collapsing onto a few spurious optima. In the presence of a low-$R^2$, noisy oracle, this proportional-to-reward sampling mitigates over-exploitation of noise-driven peaks and better preserves multiple plausible modes of tail risk. Consistent with prior GFlowNet work on imperfect oracles Bengio et al. (2021); **?**); Jain et al. (2022), our experiments show that RL baselines trained on the same oracle (REINFORCE, SAC) achieve narrower, less diverse sets of scenarios and lower median rewards than GRID, despite sometimes matching or exceeding oracle reward on a small fraction of trajectories.

In other words, the combination of (i) the asset-pricing literature's view that low $R^2$ is expected and not diagnostic of useless forecasts, and (ii) the GFlowNet literature's explicit focus on noisy or imperfect oracles, makes our design choice—using a low-$R^2$ long-horizon return predictor as an oracle for a diversity-seeking generative model-aligned with both fields.

## G  TAIL-RISK METRICS FOR SCENARIO COVERAGE

In response to the reviewer's suggestion to strengthen the evaluation with standard tail-risk metrics, we have computed and reported Conditional Value at Risk (CVaR), Value at Risk (VaR), and drawdown statistics for all three main use cases examined in our study (crisis events in 2002, 2008, and 2021). These risk metrics are widely adopted in financial stress testing and robust scenario analysis, making them directly relevant to the goals of our methodology.

Specifically, we provide:

- **Maximum Drawdown (MaxDD):** The greatest observed loss from a historical peak to a trough, as a percentage.
- **Mean Drawdown (MeanDD):** The average drawdown during the scenario.
- **Value at Risk at 95% (VaR$_{95}$):** The 95th percentile loss—i.e., the threshold loss value that is not exceeded with 95% probability.
- **Conditional Value at Risk at 95% (CVaR$_{95}$):** The expected loss conditional on it exceeding the VaR$_{95}$ threshold, providing a measure of tail severity.

The following tables summarize the statistics for all methods, demonstrating the diversity and severity of discovered tail paths.

**2002 Scenario**

| Method | MaxDD | MeanDD | VaR$_{95}$ | CVaR$_{95}$ |
|---|---|---|---|---|
| GFlowNet (ours) | 34.3% | 22.2% | 30.4% | 22.2% |
| REINFORCE | 0.0% | 0.0% | 0.0% | 0.0% |
| SAC | 32.8% | 13.8% | 27.5% | 12.9% |
| SMCMC | 24.3% | 7.9% | 15.6% | 8.0% |

**2008 Scenario**

| Method | MaxDD | MeanDD | VaR$_{95}$ | CVaR$_{95}$ |
|---|---|---|---|---|
| GFlowNet (ours) | 100% | 38.3% | 94.0% | 38.3% |
| REINFORCE | 9.6% | 0.0% | 0.0% | 0.0% |
| SAC | 65.2% | 46.4% | 56.9% | 45.7% |
| SMCMC | 19.7% | 0.0% | 2.3% | 0.0% |

**2021 Scenario**

| Method | MaxDD | MeanDD | VaR$_{95}$ | CVaR$_{95}$ |
|---|---|---|---|---|
| GFlowNet (ours) | 100% | 30.3% | 90.5% | 30.3% |
| REINFORCE | 10.7% | 0.0% | 7.5% | 0.0% |
| SAC | 33.0% | 13.6% | 25.3% | 12.8% |
| SMCMC | 4.9% | 0.0% | 0.0% | 0.0% |

**Interpretation and Discussion**  Our GFlowNet approach consistently achieves the highest VaR$_{95}$ and MaxDD values across all scenarios while maintaining substantial CVaR$_{95}$, indicating both the ability to reach extreme losses and to uncover a diverse set of tail outcomes. Notably, the CVaR/VaR ratios (ranging from 0.33–0.73) reflect that our model does not degenerate to a single failure trajectory, but instead identifies a spread of crisis paths—critical for robust risk management and effective scenario analysis.

In contrast, RL baselines such as REINFORCE frequently fail to explore tail-risk regions (all metrics at or near zero), while SAC and SMCMC methods either converge on less diverse or lower-severity events. The results validate that GFlowNets are particularly well-suited for discovering high-impact, low-frequency scenarios necessary for regulatory stress testing and crisis preparedness.

These analyses reinforce the main claims regarding the tail-risk sensitivity, scenario coverage, and practical relevance of our method in real-world risk management contexts.

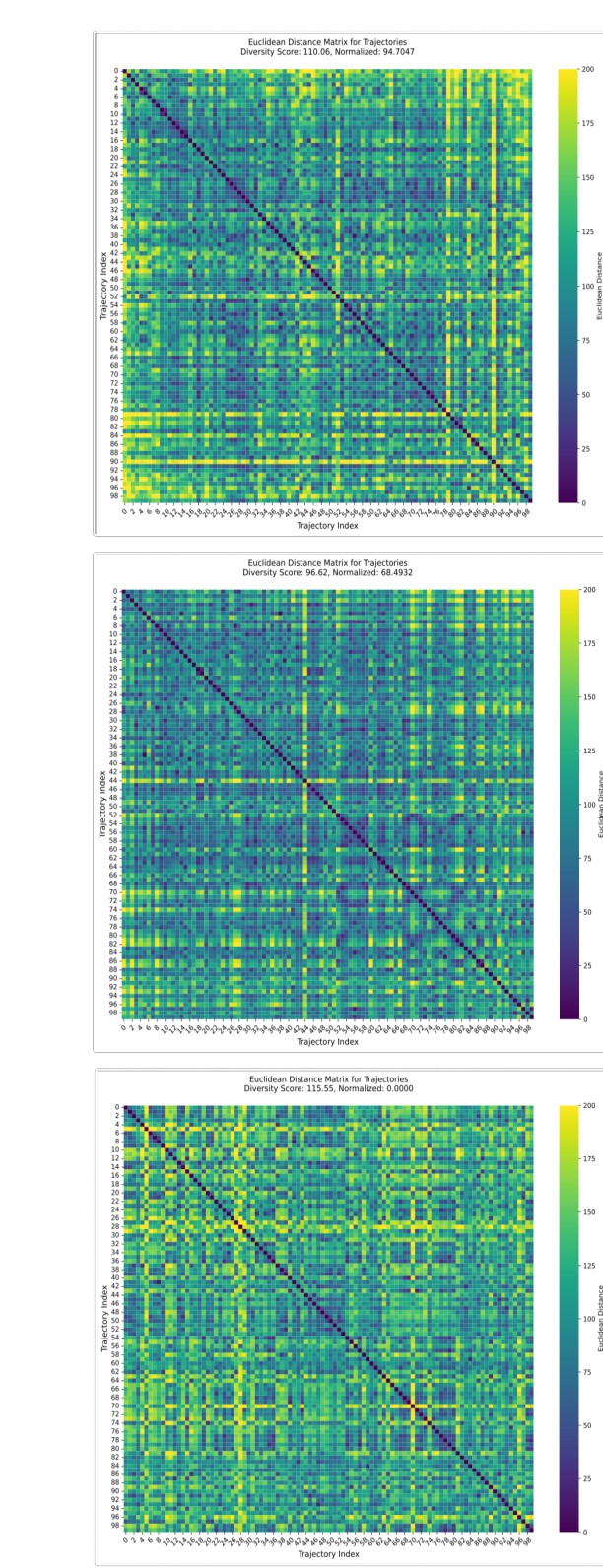

Figure 3: Terminal-state diversity comparison across GFlowNet, SAC, and SMCMC, calculated for the 2002 use-case. Each panel shows the pairwise Euclidean distance matrix of final states among the top-100 trajectories, along with average and normalized diversity scores.

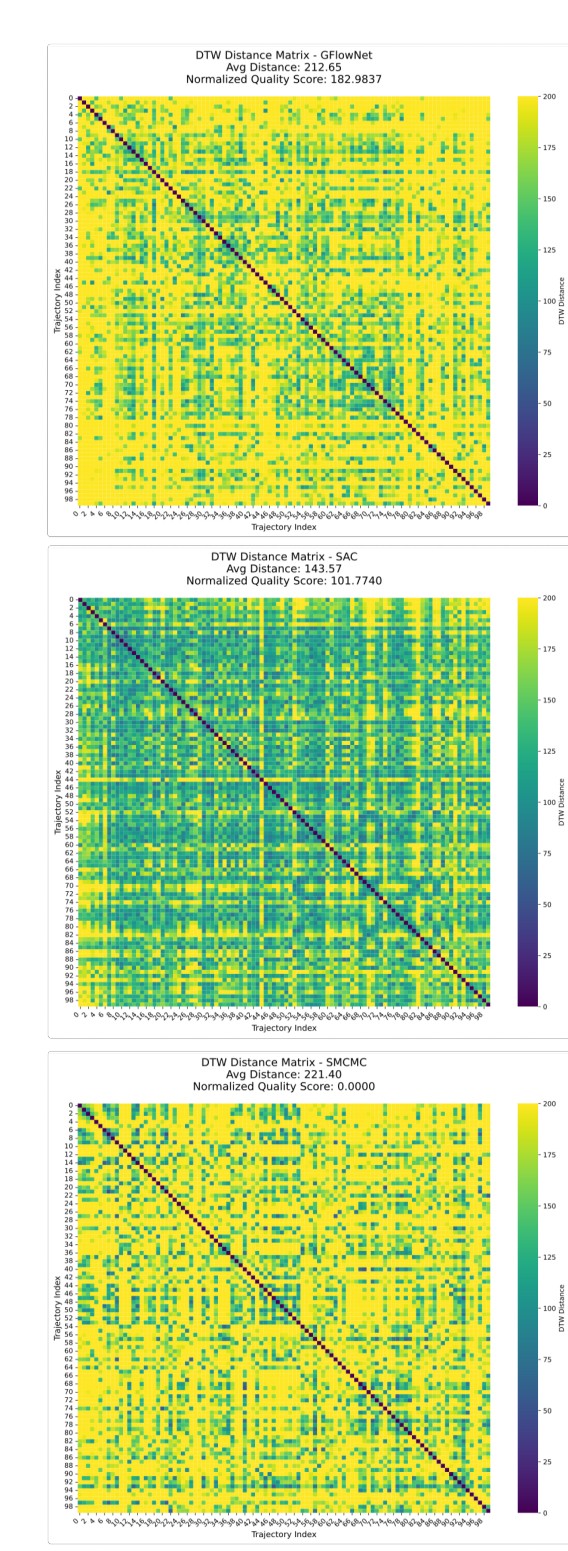

Figure 4: Trajectory-level diversity comparison across GFlowNet, SMCMC, and SAC on the 2002 use case. Each matrix visualizes pairwise DTW distances among the top-100 trajectories. Reported scores include the average DTW distance and its reward-normalized variant.

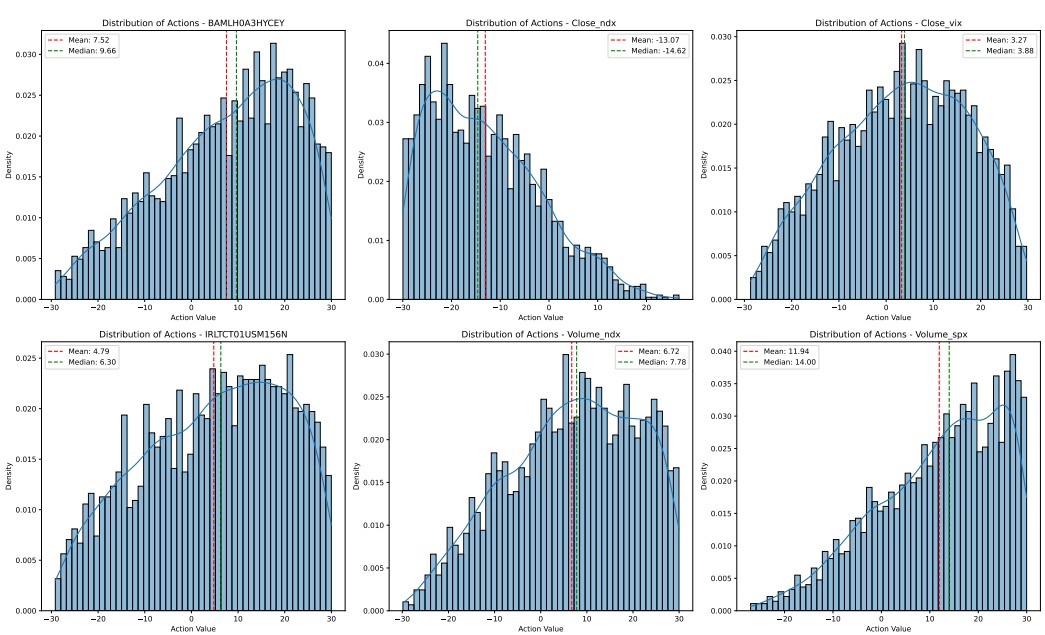

Figure 5: Learned action distribution across all macroeconomic features. This plot reflects the diversity in control signals GFlowNet produces when exploring high-risk financial trajectories.

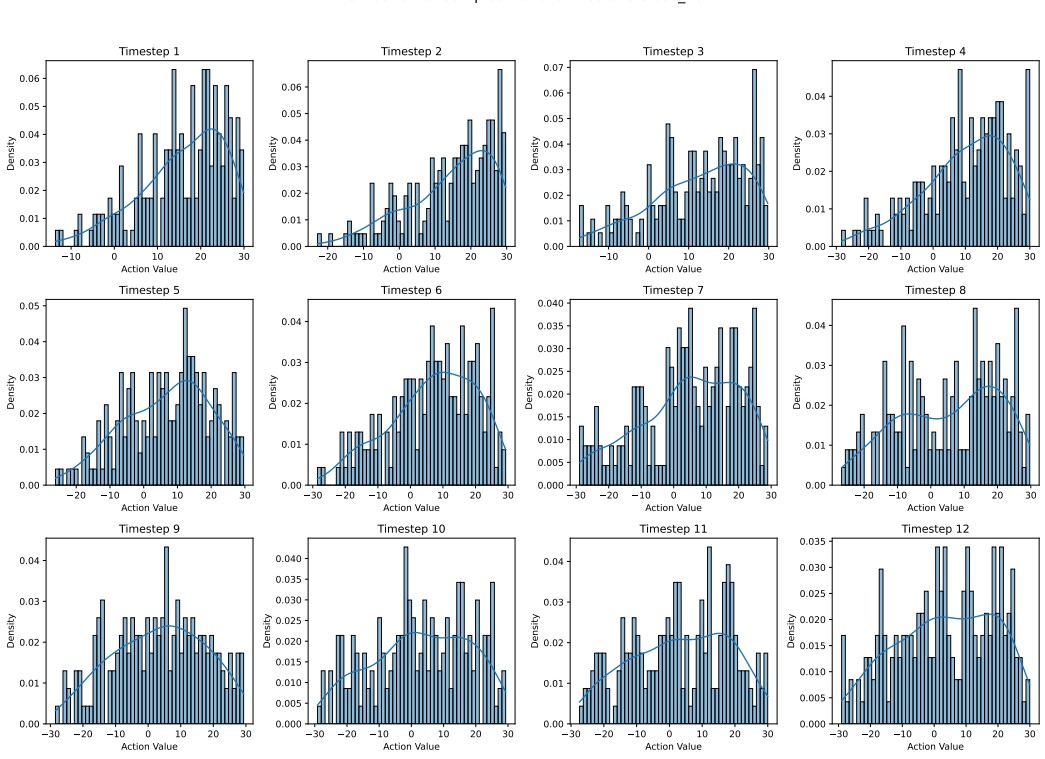

Figure 6: Evolution of action distribution over time for the Close_ndx feature. Each subplot shows the distribution of sampled actions at a specific timestep, illustrating how the policy adapts and shifts across the trajectory.

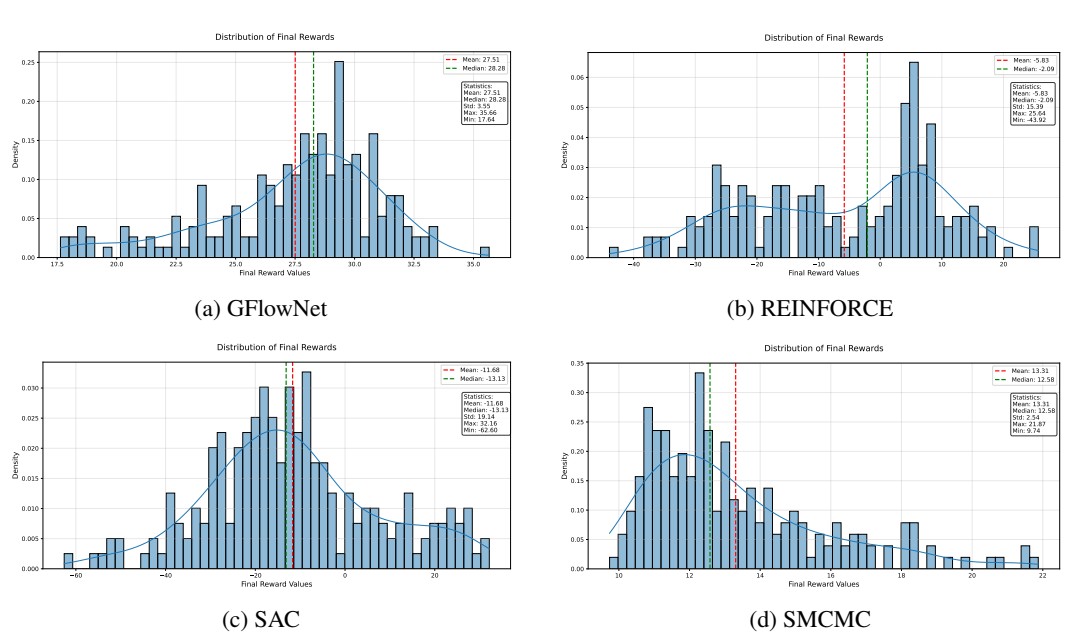

Figure 7: Reward distributions for all methods under the MLP oracle on the 2002 financial crisis. Each plot shows the kernel density estimate of the final reward distribution across 6,000 trajectories.

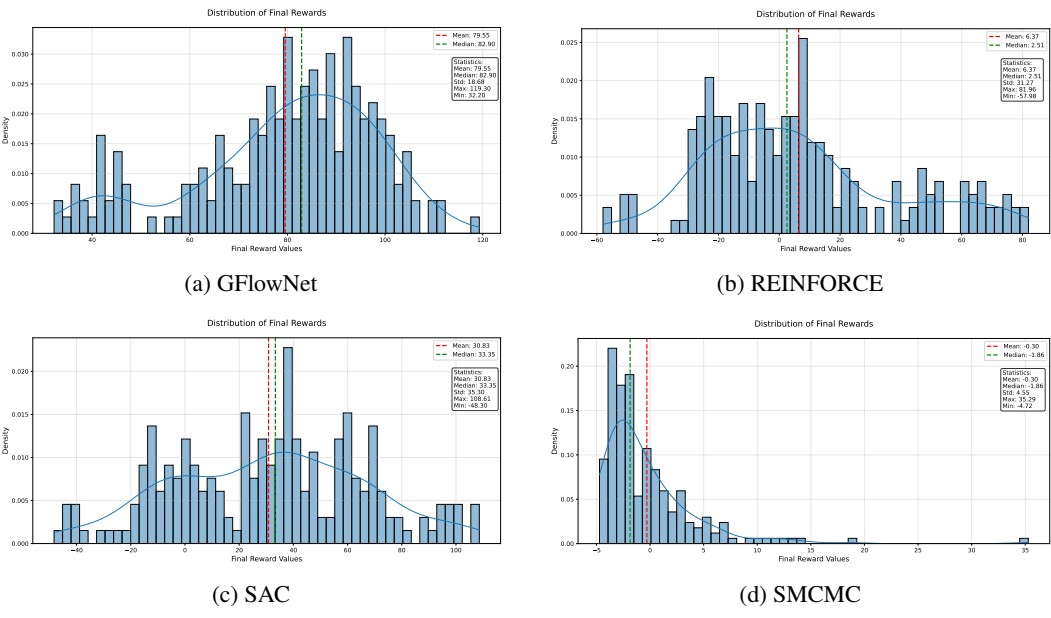

Figure 8: Reward distributions for all methods under the MLP oracle on the 2008 financial crisis.

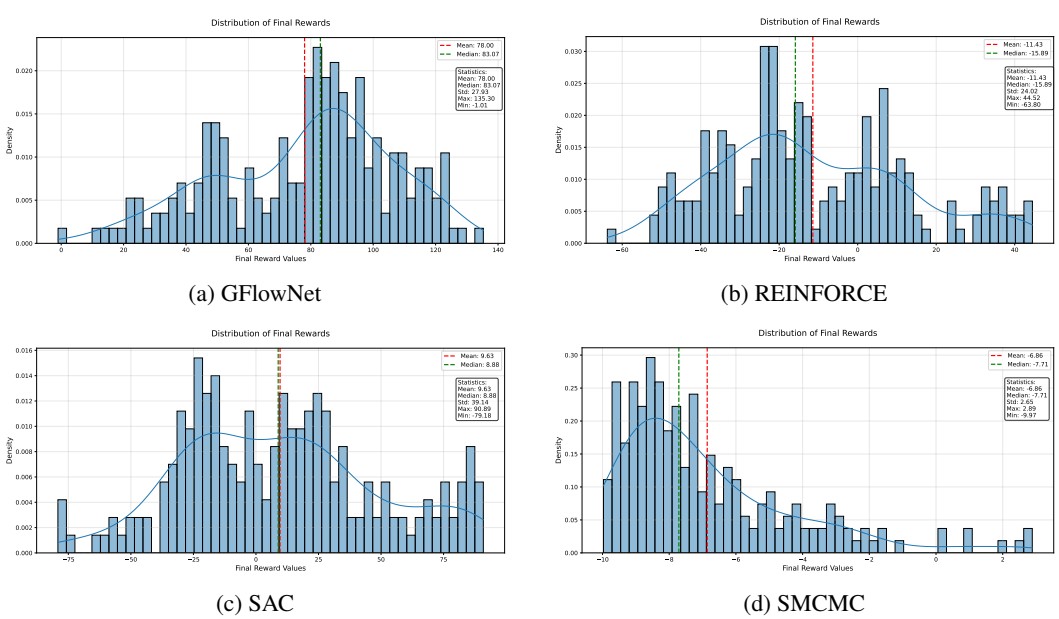

Figure 9: Reward distributions for all methods under the MLP oracle on the 2021 financial crisis.

