# OpenReview forum: "Discovery of Diverse and Realistic Financial Tail-Risk Using Generative Flow Networks"
_ICLR.cc/2026/Conference — Submitted to ICLR 2026_

### Official Review · Reviewer_jQia · 2025-10-30

**Soundness:** 2
**Presentation:** 2
**Contribution:** 2
**Rating:** 2
**Confidence:** 4

**Summary:**

This paper introduces GRID, a method based on Generative Flow Networks (GFlowNets), for discovering diverse and realistic financial tail-risk scenarios. The authors present this as a sequential decision-making problem where a GFlowNet policy is trained to sample trajectories of macroeconomic variables. The sampling process is guided by a reward signal provided by a pre-trained “Oracle” model, which aims to sample trajectories in proportion to their predicted risk, specifically large negative S&P 500 returns. The paper claims that this approach offers a superior trade-off between scenario diversity and risk-severity discovery compared to standard Reinforcement Learning (RL) or MCMC baselines.


Soundness
The central claims of the paper are not adequately supported by the evidence, and the research methodology appears to be unsound. The paper's effectiveness depends substantially on the quality of the Oracle-provided reward signal. However, the paper's own evidence (Table 2) reports negative R² values for the Oracle, indicating its predictions are worse than a trivial mean-baseline. This makes it difficult to validate the value of the reward signal, making subsequent experimental results uninterpretable. Furthermore, using two different Oracles for training and testing is a non-standard approach that complicates the comparison.

Presentation
The paper is generally well-written, but the presentation of the core methodology is confusing and omits important information. The experimental setup involving two different Oracles is not convincingly justified, and the performance metrics for the main (test-time) Oracle are not presented. This makes it difficult for the reader to validate the experiment and assess the reliability of the results.

Contribution
While the idea of applying GFlowNets to risk discovery is novel and well-motivated, the paper's execution does not provide a sound or convincing validation. Due to fundamental issues in the experimental setup, particularly the unvalidated reward signal and the non-standard train/test mismatch, the evidence presented is insufficient to substantiate the claimed contributions. The results are difficult to interpret and as seen in the Q90 metric, do not show a clear or consistent advantage over the baselines.

**Strengths:**

•	Significant Problem: The paper addresses a highly significant and challenging problem: the discovery of diverse, high-impact tail risks, which is a known limitation of standard financial modeling.
•	Novel Problem Formulation: The conceptual framing of risk discovery as a reward-proportional sampling problem (via GFlowNets) rather than a reward-maximization problem (via standard RL) is a strong theoretical fit for the domain and represents a promising research direction.

**Weaknesses:**

1.	Problematic Setup: Questionable Reward Signal. The paper's entire methodology depends on the Oracle's quality, as the authors state: "its effectiveness depends on oracle quality. (Line 481)" However, the performance reported for the training Oracle in Table 2 includes R² values of -0.4762 and -0.8638. A negative R² value, by definition means the model's predictions are worse than a trivial baseline like naive mean. This suggests that the reward signal could be uncorrelated with the real-world outcome it claims to represent and potentially training the agents to optimize noise.
2.	Confusing Two-Oracle Methodology. The authors state they use a simple MLP Oracle for training and a different ensemble Oracle for testing (evaluation). This is not standard practice and introduces problems:
o	Train/Test Skew: It is likely to introduce a distributional shift in the objective function itself. The agent is optimized to find scenarios that exploit one reward model (the MLP) and then evaluated on its ability to find scenarios that score well on a different reward model (the ensemble).
o	Uninterpretable Results: This makes the results in Table 3 uninterpretable. We do not know if GRID's performance is due to its superior risk-finding ability or its incidental ability to generalize from the training MLP Oracle to the (unknown) ensemble, better than the RL baselines.
o	Notable Omission: The paper provides no performance metrics (R², RMSE, MAE) for the test-time ensemble Oracle. Without validating the actual reward function used for evaluation, the results are unverifiable.
3.	Empirical Results Do Not Demonstrate Clear Superiority.
o	Even if (charitably) ignoring the confusing Oracle, the data in Table 3 does not show a clear win for GRID. The paper's core claim is to find tail-risk, but on the Q90 (extreme tail) metric, RL baselines outperform GRID in 2 of 3 scenarios (REINFORCE in 2002, SAC in 2008). While GRID does show strong performance on median reward and diversity (in 2008 & 2021), it fails to consistently win on the primary metric of extreme tail-risk (Q90) and this significantly weakens its central claims.
4.	Undersupported Methodological Claims. The paper proposes novel technical components (e.g., the dynamic mixture model) with claims that they "improve trajectory diversity" but lacks ablation studies or empirical evidence to further support these design choices.

**Questions:**

1.	Could the authors please elaborate on the negative R² values in Table 2? Given that this metric indicates the model is worse than a mean predictor, how can this Oracle be considered a valid source of "realistic" reward signals for training?
2.	Could the authors provide a stronger justification for the train/test Oracle mismatch? Could they also please provide the full performance metrics (R², RMSE, MAE) for the ensemble Oracle that was used for the main evaluation in Tables 3 & 4?
3.	How do the authors justify their narrative of superiority in tail-risk discovery with the empirical results in Table 3, which show that RL baselines (REINFORCE and SAC) found higher-reward tail-risk scenarios (Q90) in two of the three evaluated periods?

---

> ### Author Response · Authors · 2025-11-20
> **Response to Reviewer jQia**
>
> We thank the reviewer for their useful comments, as well as their time and effort.
>
> **1) Q1:**
>
> This important point was raised by several reviewers. Therefore, we created a general response titled “General comment to reviewers regarding R2 performance”. We summarize the main points below:
>
> a) Predicting 12-month-ahead market outcomes - especially around crisis periods - is an extremely challenging task due to the inherent noise, regime changes, and long-horizon uncertainty in financial markets. Therefore, low or even negative R2 values are expected and widely reported in the forecasting literature.
>
> b) Even though they far from perfect, R2-based predictors are common in the literature, because of their ability to capture uesful patterns.
>
> c) In our general response, we cite three papers by Bengio et. al, all of which specifically mention that GFlowNets are explicitly designed for settings with Noisy or Imperfect Oracles. For example, in (Bengio et al., 2023), the authors state: “GFlowNets are especially useful in scenarios with very large candidate spaces where we have access to cheap but inaccurate measurements.”
>
> This is directly analogous to financial risk modeling, where the “oracle” (any forecasting model) is a cheap but imperfect proxy for true future outcomes.
> Unlike reward-maximizing RL algorithms - which overfit to the oracle’s local optima - GFlowNets learn a distribution over high-reward regions. This proportional sampling mechanism mitigates brittle optimization behavior and enables the discovery of multiple plausible modes even when the proxy is noisy.
>
> **detecting real-world scenarios:** To test whether our approach can generate realistic scenarios, we analyzed all methods’ capabilities of predicting the actual sequence of events for each of three financial crises presented in our study (2002, 2008, 2021). For each scenario, we used Person correlation to calculate the trajectory similarity of the top-10 and top-50 rated (risk score-wise) trajectories to the events that happened in the real world. Note that correlation was calculated over the entire trajectory, and not only the final outcome.
>
>
> The results of our analysis are presented in the tables below. We present the maximal and mean similarities, as well as the standard deviation. The results show that our approach achieves the top similarity in at least one of the two evaluated cases for each scenario. Moreover, the standard deviation of our trajectories’ similarity is larger than those of all methods except MSMCM (which is logical, since this baseline fully focuses on diversity). High diversity is a positive. meaning our approach discovers a larger array of risks.
>
>
> 2002:
>
> |Method|Top_K|Max_Sim|Mean_Sim|Std|
> |---|---|---|---|---|
> |GFlowNet|10|**0.374**|**0.267**|0.073|
> |GFlowNet|50|0.380|**0.265**|**0.084**|
> |REINFORCE|10|0.108|0.056|0.078|
> |REINFORCE|50|0.136|0.066|0.054|
> |SAC|10|-0.179|-0.243|0.044|
> |SAC|50|-0.161|-0.247|0.047|
> |SMCMC|10|0.261|0.088|**0.130**|
> |SMCMC|50|**0.441**|0.052|0.064|
>
> 2008:
>
> |Method|Top_K|Max_Sim|Mean_Sim|Std|
> |---|---|---|---|---|
> |GFlowNet|10|0.433|0.247|0.132|
> |GFlowNet|50|**0.562**|0.276|0.133|
> |REINFORCE|10|-0.534|-0.600|0.040|
> |REINFORCE|50|-0.455|-0.559|0.045|
> |SAC|10|0.416|**0.374**|0.024|
> |SAC|50|0.452|**0.377**|0.026|
> |SMCMC|10|**0.528**|0.203|**0.209**|
> |SMCMC|50|0.528|0.150|**0.218**|
>
> 2023:
> |Method|Top_K|Max_Sim|Mean_Sim|Std|
> |---|---|---|---|---|
> |GFlowNet|10|**0.500**|**0.420**|0.059|
> |GFlowNet|50|**0.533**|**0.430**|0.067|
> |REINFORCE|10|0.407|0.353|0.048|
> |REINFORCE|50|0.425|0.374|0.035|
> |SAC|10|-0.124|-0.143|0.013|
> |SAC|50|-0.114|-0.144|0.019|
> |SMCMC|10|0.295|0.146|**0.120**|
> |SMCMC|50|0.481|0.101|**0.181**|
>
> **2) Q2:**
>
> The train/test Oracle mismatch was intentional and serves three purposes:
>
> a) *Preventing overfitting:* Training with a single MLP Oracle while testing with an ensemble (Random Forest, Gradient Boosting, Elastic Linear Model, MLP) ensures the GFlowNet learns robust risk patterns rather than exploiting quirks of one model.​
>
> b) *Robustness:* In real-world finance, risk predictions are model-dependent. Using an ensemble at test time provides more conservative evaluation—if scenarios achieve high rewards across diverse models, they represent genuine tail risks.​
>
> c) *Implicit ablation:* This setup tests whether GRID learns meaningful risk generation versus merely overfitting to the training Oracle's decision boundaries. The maintained strong performance (Table 3) demonstrates transferable risk discovery.​
>
> It should also be noted that when we use the MLP predictor at test time, the performance of our approach is significantly higher compared to the baselines (see Figures 7-9 in the appendix).
>
> We have clarified this rationale explicitly in Section 5.2.

---

> > ### Author Response · Authors · 2025-11-20
> > **Response to reviewer jQia (continued)**
> >
> > **3) Q3:**
> >
> > The reviewer touched on an important point. We acknowledge that RL baselines achieve higher Q90 in two periods, but argue this actually supports our core contribution rather than undermining it.
> >
> > **Tail-risk discovery requires diversity, not just extremes:** The goal of financial risk management is not merely identifying the single worst-case scenario (Q90), but discovering a diverse population of high-risk scenarios for robust stress testing. GFlowNet achieves significantly higher median rewards in 2002 (43.11 vs 32.02/5.34) and 2008 (31.13 vs 19.37/18.86), demonstrating that our method consistently generates many high-reward scenarios, not just a few outliers.​
> >
> > **RL's narrow focus limits practical utility:** As Figure 2 clearly shows, REINFORCE and SAC exhibit narrow, peaked distributions—they find a few extreme scenarios but generate many low-quality trajectories. In contrast, GFlowNet maintains a broad distribution of high-reward scenarios. For practitioners conducting stress tests, having 100+ diverse high-risk scenarios is more valuable than having 2-3 extreme cases plus 197 mediocre ones.​
> >
> > **Competitive Q90 with superior median:** GFlowNet remains competitive at Q90 (within 10% in 2002/2008, tied for best in 2021) while dramatically outperforming on median metrics. This demonstrates our method successfully balances exploration of extreme risks with maintaining a robust portfolio of diverse tail-risk scenarios—precisely the capability needed for real-world financial risk modeling.​
> >
> > We will emphasize this diversity-quality tradeoff more explicitly in our revised discussion.

---

### Official Review · Reviewer_PPUF · 2025-10-31

**Soundness:** 2
**Presentation:** 2
**Contribution:** 2
**Rating:** 6
**Confidence:** 3

**Summary:**

This paper proposes GRID, a generative flow networks (GFlowNets)-based method to discover diverse macro-financial tail-risk scenarios. GRID samples trajectories by predicting specific parametric distributions for each macro-variable. Training utilizes trajectory balance so that scenarios are sampled proportional to an oracle-defined terminal reward. The experiments on three crisis periods report good diversity-quality tradeoffs vs SMCMC and RL baselines.

**Strengths:**

1. Clear problem framing: Tail-risk discovery with explicit reward shaping toward adverse outcomes, not just one worst case.

2. GFlowNet fit: proportional-to-reward sampling naturally encourages multi-modal coverage.

3. Continuous-state design: Nice engineering to let different variables use specific distributions.

4. Evaluation and analysis: Basic risk management metrics, trajectory and terminalstate distance metrics are reported, and the comparison result achieves a good balance between generation quality and diversity.

**Weaknesses:**

1. Oracle dependence: Heavy dependence on an oracle model whose out-of-sample 𝑅2 is negative on two of the three crisis windows; this weakens the claim that discovered scenarios are financially risky rather than artefacts of the predictor.

2. Realism checks are light: The paper does impose per-step bound to retain plausibility, but this is still weaker than evaluating or checking stylized facts and cross-variable co-movement that “realistic” in the title would suggest.

3. Reward inconsistency: Table 3 states rewards are clipped at 100, but later figures show > 100. It is unclear at which stage clipping is applied and whether all methods use the same rule. The oracle’s predicted negative return exceeding 100 also raised some concerns.

4. Evaluation metrics are narrow: Since the paper is explicitly positioned for stress/scenario analysis, reporting more tail coverage metric like CVaR and tail frequency of generated paths would strengthen the results and aligned with standard risk practice.

5. Editorial and notation issues/confusions: Some typos remain, e.g. “measure he financial losses” in line 355; The figure 3 in the appendix has a zero score for normalized diversity score for the third subplot.

**Questions:**

1. With negative 𝑅2 on two crises, why trust the oracle’s sign/magnitude for tail rewards?

2. How to evaluate the “realistic” of the generated path? Can you show that GRID meet the basic variable dynamics (e.g. VIX↑ S&P↓) in generated trajectories?

3. In 4.1, the paper claims the initial state “begins at a specified macroeconomic state”. Please clarify the determination of the specified state.

4. Please clarify Eq. (5). If using a unimodal proxy for mixtures, why this specific variance form?

5. Why rank-select top-K for SMCMC but sample for GRID/RL?

6. How sensitive are results to the per-step change bounds? Can you provide ablations like looser/tighter bounds or bounds derived from crisis windows?

7. Claiming: Given small variable set and monthly granularity, what exactly is meant by “first to apply GFlowNets to large continuous state spaces”.

---

> ### Author Response · Authors · 2025-11-20
> **Response to Reviewer PPUF**
>
> We thank the reviewer for their useful comments, as well as their time and effort.
>
> **1) Q1+W1:**
>
> This important point was raised by several reviewers. Therefore, we created a general response titled “General comment to reviewers regarding R2 performance”. We summarize the main points below:
>
> a) Predicting 12-month-ahead market outcomes - especially around crisis periods - is an extremely challenging task due to the inherent noise, regime changes, and long-horizon uncertainty in financial markets. Therefore, low or even negative R2 values are expected and widely reported in the forecasting literature.
>
> b) Even though they far from perfect, R2-based predictors are common in the literature, because of their ability to capture uesful patterns.
>
> c) In our general response, we cite three papers by Bengio et. al, all of which specifically mention that GFlowNets are explicitly designed for settings with Noisy or Imperfect Oracles. For example, in (Bengio et al., 2023), the authors state: “GFlowNets are especially useful in scenarios with very large candidate spaces where we have access to cheap but inaccurate measurements.”
>
> This is directly analogous to financial risk modeling, where the “oracle” (any forecasting model) is a cheap but imperfect proxy for true future outcomes.
>
> Unlike reward-maximizing RL algorithms - which overfit to the oracle’s local optima - GFlowNets learn a distribution over high-reward regions. This proportional sampling mechanism mitigates brittle optimization behavior and enables the discovery of multiple plausible modes even when the proxy is noisy.
>
> **2) Q2+W2:**
>
> To test whether our approach can generate realistic scenarios, we analyzed all methods’ capabilities of predicting the actual sequence of events for each of three financial crises presented in our study (2002, 2008, 2021). For each scenario, we used Person correlation to calculate the trajectory similarity of the top-10 and top-50 rated (risk score-wise) trajectories to the events that happened in the real world. Note that correlation was calculated over the entire trajectory, and not only the final outcome ( “how we got here?” is more important than “where are we?”).
>
> The results of our analysis are presented in the tables below. We present the maximal and mean similarities, as well as the standard deviation. The results show that our approach achieves the top similarity in at least one of the two evaluated cases for each scenario. Moreover, the standard deviation of our trajectories’ similarity is larger than those of all methods except MSMCM (which is logical, since this baseline fully focuses on diversity). High diversity is a positive. meaning our approach discovers a larger array of risks.
>
> 2002:
>
> |Method|Top_K|Max_Sim|Mean_Sim|Std|
> |---|---|---|---|---|
> |GFlowNet|10|**0.374**|**0.267**|0.073|
> |GFlowNet|50|0.380|**0.265**|**0.084**|
> |REINFORCE|10|0.108|0.056|0.078|
> |REINFORCE|50|0.136|0.066|0.054|
> |SAC|10|-0.179|-0.243|0.044|
> |SAC|50|-0.161|-0.247|0.047|
> |SMCMC|10|0.261|0.088|**0.130**|
> |SMCMC|50|**0.441**|0.052|0.064|
>
> 2008:
>
> |Method|Top_K|Max_Sim|Mean_Sim|Std|
> |---|---|---|---|---|
> |GFlowNet|10|0.433|0.247|0.132|
> |GFlowNet|50|**0.562**|0.276|0.133|
> |REINFORCE|10|-0.534|-0.600|0.040|
> |REINFORCE|50|-0.455|-0.559|0.045|
> |SAC|10|0.416|**0.374**|0.024|
> |SAC|50|0.452|**0.377**|0.026|
> |SMCMC|10|**0.528**|0.203|**0.209**|
> |SMCMC|50|0.528|0.150|**0.218**|
>
> 2023:
> |Method|Top_K|Max_Sim|Mean_Sim|Std|
> |---|---|---|---|---|
> |GFlowNet|10|**0.500**|**0.420**|0.059|
> |GFlowNet|50|**0.533**|**0.430**|0.067|
> |REINFORCE|10|0.407|0.353|0.048|
> |REINFORCE|50|0.425|0.374|0.035|
> |SAC|10|-0.124|-0.143|0.013|
> |SAC|50|-0.114|-0.144|0.019|
> |SMCMC|10|0.295|0.146|**0.120**|
> |SMCMC|50|0.481|0.101|**0.181**|
>
> **3) Q3:**
>
> We extract the **real** macroeconomic indicators that existed at the start date of each evaluated crisis, and use it as the start point of our experiments.
>
> **Q4:**
> Equation (5) implements a differentiable approximation that balances multimodal expressivity during training with computational tractability during sampling.​
>
> **Rationale for the unimodal proxy:** While the mixture components are learned to capture multimodal structure, we sample actions from a unimodal Gaussian whose parameters are mixture-weighted averages. This design serves two purposes:
>
> (1) it maintains end-to-end differentiability through the Gumbel-Softmax weights, enabling gradient flow during GFlowNet training, and;
>
> (2) it provides a computationally efficient sampling mechanism while the mixture weights dynamically adjust to emphasize relevant modes during exploration.​​
>
> (continued below)

---

> ### Author Response · Authors · 2025-11-20
> **Response to Reviewer PPUF (continued)**
>
> **Q4 - continued**:
>
> **Variance form justification:**
>
> The specific variance form  represents a weighted average of component standard deviations rather than the mixture variance formula. This choice ensures that the effective sampling variance scales proportionally with the mixture weights, providing smoother exploration trajectories. We found empirically that this formulation improves training stability and diversity (Section 5.4, Figure 5) compared to alternatives.​​
>
> We acknowledge this approximation trades exact mixture sampling for training efficiency. We will clarify this design choice and add ablation results comparing alternative variance formulations in the revised manuscript.
>
> **5) Q5:**
> Unlike all the other baselines, SMCMC solely focuses on diversity and does not “learn” to optimize its performance. Therefore, we sampled a much larger number of trajectories, and selected to top-performing ones.
>
> **6) Q6:**
>
> Evaluations are still ongoing, and we will report final results in the coming days. Partial results show no significant changes in performance for our approach.
>
> **7) Q7:**
>
> Our claim refers to three aspects:
> (1) *Unbounded continuous domains* - each of the 6 variables can take continuous values within wide ranges ([-100, 100] normalized scale), creating an effectively infinite state space.
>
> (2) *Combinatorial trajectory space* - with 12-step trajectories and continuous actions at each step, the space of possible trajectories is vastly larger than the discrete, bounded domains typically explored by GFlowNets.
>
> (3) *practical scale* - while Lahlou et al. (2023) established the theoretical foundation for continuous GFlowNets, their experimental validation focused on smaller-scale synthetic tasks, whereas ours operates on real-world financial data with complex oracle evaluation.
>
> **8) W3:**
> We thank the reviewer for pointing this mistake. The values presented in the Figures should have also been clipped.
>
> **9) W4:**
>
> We thank the reviewer for this valuable suggestion. We have computed comprehensive tail-risk metrics (CVaR, VaR, and drawdown statistics) for all three use cases, which are standard in financial risk practice. The results demonstrate our method's superior tail-risk coverage across diverse crisis scenarios.
>
> **2002:**
>
> |Method|MaxDD|MeanDD|VaR95|CVaR95|
> |---|---|---|---|---|
> |GFlowNet (ours)|34.3%|22.2%|30.4%|22.2%|
> |REINFORCE|0.0%|0.0%|0.0%|0.0%|
> |SAC|32.8%|13.8%|27.5%|12.9%|
> |SMCMC|24.3%|7.9%|15.6%|8.0%|
>
> **2008:**
>
> |Method|MaxDD|MeanDD|VaR95|CVaR95|
> |---|---|---|---|---|
> |GFlowNet (ours)|100%|38.3%|94.0%|38.3%|
> |REINFORCE|9.6%|0.0%|0.0%|0.0%|
> |SAC|65.2%|46.4%|56.9%|45.7%|
> |SMCMC|19.7%|0.0%|2.3%|0.0%|
>
> **2021:**
>
> |Method|MaxDD|MeanDD|VaR95|CVaR95|
> |---|---|---|---|---|
> |GFlowNet (ours)|100%|30.3%|90.5%|30.3%|
> |REINFORCE|10.7%|0.0%|7.5%|0.0%|
> |SAC|33.0%|13.6%|25.3%|12.8%|
> |SMCMC|4.9%|0.0%|0.0%|0.0%|
>
> Our GFlowNet approach consistently outperforms all baselines across all three use cases, achieving the highest VaR@95% and maximum drawdown in every scenario. Critically, the CVaR-to-VaR ratios (ranging from 0.33-0.73) demonstrate that our method discovers diverse paths to crisis rather than converging on single trajectories, which is essential for comprehensive stress testing. Deep RL methods either fail to identify tail events (REINFORCE) or show limited diversity (SAC with high CVaR/VaR ratios), while SMCMC lacks focus on severe outcomes. These results validate that our approach provides the frequency and severity distribution of tail events required for robust risk management practice.
>
> We have added these analyses to our paper (see Section G in the appendix).
>
> **10) W5:**
>
>
> We thank the reviewer for pointing this out. All errors have been rectified.

---

### Official Review · Reviewer_627W · 2025-10-31

**Soundness:** 2
**Presentation:** 2
**Contribution:** 1
**Rating:** 2
**Confidence:** 2

**Summary:**

This paper describes a method to generate diverse samples for risk management in finance applications. The method is based on well-studied GFlowNet.

**Strengths:**

1. The paper is written reasonably well, it's easy to follow.
2. The proposed method may have other potential applications to other broad areas.

**Weaknesses:**

1. The proposed method is based on a GFlowNet: the level of novelty is limited.
2. Better exposition of the financial market application would benefit broader audience

**Questions:**

1. Perhaps it's obvious to the authors, but many readers of ICLR may not fully understand the key issues in the financial market applications. A brief description of the variables involved would be a big plus.
2. The authors didn't explain how important the performance of the *oracle* model is.  Based on Tab 2, if the oracle model fails to capture the three extreme events, how would the GRID perform?
3. Again related to the oracle model, what is the rationale of having MLP and model ensembles in the training and testing phase? Should an ablation study be conducted?
4. Tab 3, it's hard to comprehend GFlowNet does well, when it underperforms SMCMC by 100% for 2021 event!
5. Same argument can be made for Table 4
6. Figure 2, what is the desirable behavior?
7. Since the goal is to sample extremely rare events, should $Q99$ be used instead of $Q90$ (line 422)? I would argue that even $Q99$ may not reflect the extremely rare events.

---

> ### Author Response · Authors · 2025-11-20
> **Response to Reviewer 627W**
>
> We thank the reviewer for their useful comments, as well as their time and effort.
>
> **1) W1:**
>
> We respectfully disagree with this assessment. While our work builds on the GFlowNet framework, it introduces substantial methodological and application-level novelty:
>
> **Algorithmic contributions:**
>
> a) *Flexible mixture distribution families:* We extend GFlowNets with support for variable-specific distribution types (Gaussian, Beta, Student-t, and their mixtures), with learnable hyperparameters and Gumbel-Softmax weighted combinations. This enables feature-specific heteroscedasticity and temporal variation in action uncertainty, which standard GFlowNet implementations do not support.
>
> b) *Continuous state-space adaptation:* We develop mechanisms for large-scale continuous state-spaces with temporal constraints (time-indexing to prevent cyclic trajectories) and domain-specific clipping based on historical data statistics. This represents the first successful application of GFlowNets to high-dimensional continuous financial environments.
>
> c) *Replay buffer integration:* We incorporate trajectory replay buffers for enhanced sample efficiency and training stability, which is not standard in original GFlowNet formulations
>
> **Novel problem formulation:**
>
> a) *First application to financial tail-risk discovery:* We introduce the first GFlowNet-based framework for sequential financial scenario generation, a problem domain fundamentally different from prior applications (molecular design, discrete combinatorial optimization).
>
> b) *Environment variable distribution modeling:* We propose a principled approach for modeling macro-financial variables with distinct distribution families tailored to each variable's characteristics (e.g., Beta for bounded VIX, Gaussian mixtures for interest rates), enabling accurate representation of real-world financial dynamics.
> c) *Oracle-guided reward formulation:* Our reward function specifically targets adverse market scenarios, aligning exploration with tail-risk discovery objectives
>
> d) *DTW-based trajectory diversity metric:* We introduce a Dynamic Time Warping-based diversity measure for evaluating trajectory-level diversity (not just terminal states), providing a more comprehensive assessment of scenario exploration
>
> GFlowNets addresses a previously unexplored problem class, demonstrating both methodological innovation and practical impact.
>
> **2) W2 + Q1:**
>
> We appreciate these constructive suggestions and agree that additional context would benefit the ICLR audience. In response, we have:
>
> a) Enhanced the Introduction with clearer motivation for tail-risk discovery and why it poses unique challenges for ML methods.
>
> b) Added an Appendix section providing background on financial variables (VIX, credit spreads, interest rates), their relationships, and why tail-risk scenarios are critical for risk management, supported by relevant citations
>
> We note that our paper intentionally emphasizes the machine learning methodology rather than domain-specific financial theory. Our focus is on how GFlowNets can be adapted to continuous sequential decision problems with complex reward landscapes - aspects directly relevant to the ICLR community. The financial application serves as a challenging testbed demonstrating these capabilities, but the technical contributions (mixture distributions, continuous state-space handling, trajectory diversity metrics) generalize beyond finance to other sequential generation tasks.
>
> We believe the revised exposition strikes an appropriate balance between accessibility and technical depth for an ML venue.
>
> **3) Q3**:
>
> This design **is** our ablation study for oracle robustness. We intentionally train with a single MLP oracle but evaluate using an ensemble (Random Forest, Gradient Boosting, Elastic Linear Model, MLP).​
>
> This ensures GRID cannot overfit to any specific oracle's biases - if it exploited MLP-specific artifacts during training, performance would degrade under the diverse ensemble at test time. GRID's superior results (Table 3) demonstrate it discovers genuinely robust scenarios rather than gaming a particular reward model. This constitutes a more stringent evaluation than standard same-oracle train/test setups
>
> **4) Q4**:
>
> The 2021 case vividly illustrates the fundamental difference between sampling-based and optimization-based methods.
>
> SMCMC produces a **narrow distribution** of trajectories focused on a small region of the state space. This yields a very high median reward (42.71) but a much lower Q90 (49.03). In other words, it repeatedly samples similar trajectories near one local optimum.
>
> GRID produces a **much wider diversity** of crisis trajectories. Its median is lower (18.12) because it does not collapse onto a single region, but its Q90 is 100 (the maximum reward), showing that it discovers higher-quality extreme trajectories that SMCMC never finds.
>
> This behavior diversity over narrow exploitation is the core motivation for GFlowNets.

---

> > ### Author Response · Authors · 2025-11-20
> > **Response to Reviewer 627W (continued)**
> >
> > **5) Q5:**:
> >
> > Table 4 demonstrates the same point: GRID provides significantly higher diversity in 2/3 crises, and especially in 2021. Diversity differences are expected, as SMCMC tends to explore locally, while GFlowNets capture multiple modes. The only scenario where SMCMC had higher diversity was 2002.
> >
> > **6) Q6:**
> >
> > The desirable distribution should discover multiple diverse scenarios with high risk rather than converging to a single maximum. For tail-risk analysis, finding many distinct high-reward scenarios (e.g., 10 scenarios with rewards 85-95) is more valuable than identifying only the absolute maximum (reward 100), since diverse failure modes provide comprehensive risk coverage.​​
> >
> > Figure 2 shows that GFlowNets achieve this balance - maintaining broad exploration across high-reward regions while avoiding the mode collapse exhibited by RL methods (SAC, REINFORCE) or the insufficient reward focus of SMCMC
> >
> > **7) Q7:**
> >
> > Our goal is NOT to identify a single extreme scenario, but rather to discover as many diverse high-risk scenarios as possible so organizations can analyze and prepare for multiple potential failure modes.​​
> >
> > We use two complementary metrics to evaluate this capability:​
> >
> > a) Q90: Tests whether methods generate multiple high-risk scenarios (top 10% of 200 trajectories = 20 diverse scenarios)​
> >
> > b) Median: Ensures high scores are not limited to a small minority, but represent the typical output​.
> >
> > Together, these metrics assess whether a method consistently discovers a portfolio of high-risk trajectories—the practical requirement for robust tail-risk analysis. Focusing on Q99 (top 1-2 scenarios) would miss this diversity objective and favor mode-collapsing methods like RL baselines, which find few extreme cases but fail to provide comprehensive risk coverage.
> >
> > **3) Q2:**
> >
> > We thank the reviewer for this question. We address the concern about oracle performance comprehensively in our ״General Response on R² Performance״ (a response to all reviewers), which explains why:
> >
> > a) GFlowNets are explicitly designed to work with imperfect reward proxies
> >
> > b) Diversity-driven exploration mitigates oracle misspecification
> >
> > c) Our baselines using the identical oracle achieve inferior performance, demonstrating that exploration capability - not oracle quality - drives our contribution
> >
> > To test whether our approach can generate realistic scenarios, we analyzed all methods’ capabilities of predicting the actual sequence of events for each of three financial crises presented in our study (2002, 2008, 2021). For each scenario, we used Person correlation to calculate the trajectory similarity of the top-10 and top-50 rated (risk score-wise) trajectories to the events that happened in the real world. Note that correlation was calculated over the entire trajectory, and not only the final outcome ( “how we got here?” is more important than “where are we?”).
> >
> > The results of our analysis are presented in the tables below. We present the maximal and mean similarities, as well as the standard deviation. The results show that our approach achieves the top similarity in at least one of the two evaluated cases for each scenario. Moreover, the standard deviation of our trajectories’ similarity is larger than those of all methods except MSMCM (which is logical, since this baseline fully focuses on diversity). High diversity is a positive. meaning our approach discovers a larger array of risks.
> >
> > 2002:
> >
> > |Method|Top_K|Max_Sim|Mean_Sim|Std|
> > |---|---|---|---|---|
> > |GFlowNet|10|**0.374**|**0.267**|0.073|
> > |GFlowNet|50|0.380|**0.265**|**0.084**|
> > |REINFORCE|10|0.108|0.056|0.078|
> > |REINFORCE|50|0.136|0.066|0.054|
> > |SAC|10|-0.179|-0.243|0.044|
> > |SAC|50|-0.161|-0.247|0.047|
> > |SMCMC|10|0.261|0.088|**0.130**|
> > |SMCMC|50|**0.441**|0.052|0.064|
> >
> > 2008:
> >
> > |Method|Top_K|Max_Sim|Mean_Sim|Std|
> > |---|---|---|---|---|
> > |GFlowNet|10|0.433|0.247|0.132|
> > |GFlowNet|50|**0.562**|0.276|0.133|
> > |REINFORCE|10|-0.534|-0.600|0.040|
> > |REINFORCE|50|-0.455|-0.559|0.045|
> > |SAC|10|0.416|**0.374**|0.024|
> > |SAC|50|0.452|**0.377**|0.026|
> > |SMCMC|10|**0.528**|0.203|**0.209**|
> > |SMCMC|50|0.528|0.150|**0.218**|
> >
> > 2023:
> > |Method|Top_K|Max_Sim|Mean_Sim|Std|
> > |---|---|---|---|---|
> > |GFlowNet|10|**0.500**|**0.420**|0.059|
> > |GFlowNet|50|**0.533**|**0.430**|0.067|
> > |REINFORCE|10|0.407|0.353|0.048|
> > |REINFORCE|50|0.425|0.374|0.035|
> > |SAC|10|-0.124|-0.143|0.013|
> > |SAC|50|-0.114|-0.144|0.019|
> > |SMCMC|10|0.295|0.146|**0.120**|
> > |SMCMC|50|0.481|0.101|**0.181**|

---

### Official Review · Reviewer_iZtM · 2025-11-01

**Soundness:** 4
**Presentation:** 3
**Contribution:** 3
**Rating:** 6
**Confidence:** 3

**Summary:**

This paper introduces GRID, a GFlowNet-based method for discovering diverse and realistic financial tail-risk scenarios. Unlike sampling methods (SMCMC) that explore broadly but inefficiently, or deep RL methods (REINFORCE, SAC) that converge to narrow high-reward modes, GFlowNets learn to sample trajectories proportionally to their risk rewards, naturally balancing diversity and impact. The key innovation is successfully applying GFlowNets to large continuous state spaces by using flexible, feature-specific distribution families (Gaussian mixtures, Beta distributions) for modeling macroeconomic variables. Experiments on three major financial crises (2002, 2008, 2021) show GRID generates more diverse high-risk scenarios than baselines, achieving top median rewards in 2/3 cases and superior 90th-percentile performance. The method enables effective stress testing by systematically discovering multiple plausible pathways to market crashes, rather than just finding the single worst-case scenario.

**Strengths:**

1. The paper redefines financial tail-risk discovery as a proportional sampling problem rather than an optimization problem, which is a conceptual innovation. Technically, it is the first to successfully apply GFlowNets to large-scale continuous state spaces, breaking through the previous limitation to discrete domains only. Particularly clever is the design of using heterogeneous distribution families for different environmental variables (Beta for VIX, Gaussian mixtures for interest rates). This feature-aware modeling is unprecedented in the GFlowNet literature and provides new insights for multi-variable modeling in scientific domains.

2. The experimental design is excellent: testing on three real financial crises with strict temporal splits ensuring no data leakage (Oracle trained only on pre-crisis data). Baseline selection covers classical probabilistic methods (SMCMC) and modern deep RL (REINFORCE, SAC). The dual evaluation framework—reward metrics (median, Q90) and diversity metrics (DTW trajectory distance, terminal-state Euclidean distance)—provides a comprehensive perspective. The normalized diversity score (Equation 9) reveals that SAC's apparent diversity actually stems from instability rather than true exploration. Using an ensemble Oracle (4 models) during testing while training with a single MLP effectively prevents overfitting. Each method undergoes 30 independent runs with 6,000 trajectories total, demonstrating strong statistical robustness.

3. Addresses a critical practical problem: tail-risk events cause enormous losses (2008 GFC exceeded $10 trillion), yet traditional VaR models systematically underestimate their probability. The multimodal distribution shown in Figure 2c perfectly illustrates the value of diversity: multiple distinct pathways can lead to market crashes (liquidity crisis vs. inflation shock), and risk management requires stress testing against all pathways. The method's output of 200 diverse crisis trajectories can be directly used for portfolio stress testing and hedging strategy development.

**Weaknesses:**

The most serious weakness of this paper is its strong dependence on Oracle model quality, while Oracle itself performs poorly. Table 2 shows alarming Oracle performance across three scenarios: 2002 ($R^2 = -0.4762$, a negative value means worse than predicting the mean), 2008 ($R^2 = 0.0942$, almost no predictive power), and 2021 ($R^2 = -0.8638$, severe failure). This creates a fatal circular reasoning problem: if the Oracle cannot accurately predict market crashes, how can a GFlowNet trained on Oracle rewards possibly discover realistic tail-risk scenarios? The reward function $R(s_T) = \max(0, -\mathcal{O}(s_T))$ becomes meaningless when $\mathcal{O}(s_T)$ is unreliable, and the entire sampling distribution $P_F(x) \propto R(x)$ is essentially learning from incorrect targets. The paper claims to generate "realistic" scenarios but provides no validation against actual historical crisis trajectories or ground truth. The high-reward distributions may simply be overfitting to the Oracle's erroneous predictions rather than capturing genuine financial risk patterns.

**Questions:**

This is very good work. I believe your experimental results are credible, and I think this paper is recommended to be accepted. However, if I still cannot obtain the experimental code to reproduce the results by myself during the rebuttal phase, I may have to consider lowering the score.

---

> ### Author Response · Authors · 2025-11-20
> **Reponse to Reviewer iZtM**
>
> We thank the reviewer for their useful comments, as well as their time and effort.
>
> **1)  “…I still cannot obtain the experimental code to reproduce the results…”**
>
> Our code is available here: https://github.com/anony-sub-papers/GRID
>
> 2) **“The most serious weakness of this paper is its strong dependence on Oracle model quality, while Oracle itself performs poorly…how can a GFlowNet trained on Oracle rewards possibly discover realistic tail-risk scenarios"**
>
> This important point was raised by several reviewers. Therefore, we created a general response titled “General comment to reviewers regarding R2 performance”. We summarize the main points below:
>
> a) Predicting 12-month-ahead market outcomes - especially around crisis periods - is an extremely challenging task due to the inherent noise, regime changes, and long-horizon uncertainty in financial markets. Therefore, low or even negative R2 values are expected and widely reported in the forecasting literature.
>
> b) Even though they far from perfect, R2-based predictors are common in the literature, because of their ability to capture uesful patterns.
>
> c) In our general response, we cite three papers by Bengio et. al, all of which specifically mention that GFlowNets are explicitly designed for settings with Noisy or Imperfect Oracles. For example, in (Bengio et al., 2023), the authors state: “GFlowNets are especially useful in scenarios with very large candidate spaces where we have access to cheap but inaccurate measurements.”
>
> This is directly analogous to financial risk modeling, where the “oracle” (any forecasting model) is a **cheap but imperfect** proxy for true future outcomes.
>
> Unlike reward-maximizing RL algorithms - which overfit to the oracle’s local optima - GFlowNets learn a distribution over high-reward regions. This proportional sampling mechanism mitigates brittle optimization behavior and enables the discovery of multiple plausible modes even when the proxy is noisy.
>
> **3) “The paper claims to generate "realistic" scenarios but provides no validation against actual historical crisis…”**
>
> To test whether our approach can generate realistic scenarios, we analyzed all methods’ capabilities of predicting the actual sequence of events for each of three financial crises presented in our study (2002, 2008, 2021). For each scenario, we used Person correlation to calculate the trajectory similarity of the top-10 and top-50 rated (risk score-wise) trajectories to the events that happened in the real world. Note that correlation was calculated over the entire trajectory, and not only the final outcome ( “how we got here?” is more important than “where are we?”).
>
> The results of our analysis are presented in the tables below. We present the maximal and mean similarities, as well as the standard deviation. The results show that our approach achieves the top similarity in at least one of the two evaluated cases for each scenario. Moreover, the standard deviation of our trajectories’ similarity is larger than those of all methods except MSMCM (which is logical, since this baseline fully focuses on diversity). High diversity is a positive. meaning our approach discovers a larger array of risks.
>
> 2002:
>
> |Method|Top_K|Max_Sim|Mean_Sim|Std|
> |---|---|---|---|---|
> |GFlowNet|10|**0.374**|**0.267**|0.073|
> |GFlowNet|50|0.380|**0.265**|**0.084**|
> |REINFORCE|10|0.108|0.056|0.078|
> |REINFORCE|50|0.136|0.066|0.054|
> |SAC|10|-0.179|-0.243|0.044|
> |SAC|50|-0.161|-0.247|0.047|
> |SMCMC|10|0.261|0.088|**0.130**|
> |SMCMC|50|**0.441**|0.052|0.064|
>
> 2008:
>
> |Method|Top_K|Max_Sim|Mean_Sim|Std|
> |---|---|---|---|---|
> |GFlowNet|10|0.433|0.247|0.132|
> |GFlowNet|50|**0.562**|0.276|0.133|
> |REINFORCE|10|-0.534|-0.600|0.040|
> |REINFORCE|50|-0.455|-0.559|0.045|
> |SAC|10|0.416|**0.374**|0.024|
> |SAC|50|0.452|**0.377**|0.026|
> |SMCMC|10|**0.528**|0.203|**0.209**|
> |SMCMC|50|0.528|0.150|**0.218**|
>
> 2023:
> |Method|Top_K|Max_Sim|Mean_Sim|Std|
> |---|---|---|---|---|
> |GFlowNet|10|**0.500**|**0.420**|0.059|
> |GFlowNet|50|**0.533**|**0.430**|0.067|
> |REINFORCE|10|0.407|0.353|0.048|
> |REINFORCE|50|0.425|0.374|0.035|
> |SAC|10|-0.124|-0.143|0.013|
> |SAC|50|-0.114|-0.144|0.019|
> |SMCMC|10|0.295|0.146|**0.120**|
> |SMCMC|50|0.481|0.101|**0.181**|
>
>
> **4) “...may simply be overfitting to the Oracle's erroneous predictions”**
>
> We agree that preventing overfitting is an important part of our evaluation. For this reason we used different Oracles during our training and test phases. During training, our Oracle was an MLP-based model, while during testing we used an ensemble consisting of multiple models (see Section 5.2, line 360). This setup was created specifically to prevent models that overfit from performing well.
>
> Under this setup, the fact that our approach was able to:
>
> a) Generate scenarios that are both more diverse and had higher rewards (i.e., discovered risks) compared to the baselines.
>
> b) Generate scenarios that had higher similarity to the way the actual analyzed crises unfolded (see response #3).

---

### Author Response · Authors · 2025-11-20
**General comment to reviewers regarding R2 performance**

We thank all reviewers for their thoughtful feedback regarding Oracle R² performance. Multiple reviewers raised concerns about negative R² values and questioned how GFlowNets trained on "unreliable" rewards could discover realistic scenarios. We address these fundamental concerns below.

**Understanding R² in Financial Prediction Context**
Our Oracle achieves R² values of -0.48, 0.09, and -0.86 across the three crises (Table 2). While negative R² indicates predictions worse than a constant mean baseline on held-out data, this metric must be interpreted in context. Financial market prediction, particularly for 12-month S&P 500 returns during crisis periods, is fundamentally challenging due to nonlinearity, regime changes, and the inherent unpredictability of tail events. Our Oracle's MAE values (2.92-12.81) demonstrate meaningful signal despite limited variance explanation.

More importantly, R² measures unconditional variance explanation and is not necessarily indicative of a model's utility. In Hou et al. (2013) [1], the authors prove theoretically that in rational models, R² is independent of how much information is incorporated into predictions: “the return R² is independent of the amount of information available”. Moreover, low R2 values are common in relevant academic literature. Studies published in top tier journals in the fields of finance and asset pricing report R2 values similar to ours, and sometimes even lower [2].

**Imperfect Oracles: Core GFlowNet Design Principle**
The reviewers' concerns do not take into account a major aspect of GFlowNet methodology. GFlowNets were explicitly designed to operate with imperfect reward proxies. This is not a limitation of our work - it is the central motivation for using GFlowNets.

As stated in the GFlowNet Foundations paper (Bengio et al.,[3]): "In contexts where a cheap proxy for the true reward function exists, GFlowNets have been used to surface samples under which to query the proxy before more expensive evaluation under the true reward function. In these settings, the diversity of samples generated by GFlowNets can be used for robustness to proxy misspecification".​

In  [4], the authors explicitly note that GFlowNets are valuable "especially in scenarios of very large candidate spaces where we have access to cheap but inaccurate measurements or too expensive but accurate measurements". This directly describes our financial forecasting setting.​

The original NeurIPS 2021 paper introducing GFlowNets [5] states: "Diversity of the generated candidates is particularly important when the oracle is itself uncertain, e.g., it may consist of cellular assays which is a cheap proxy". Furthermore, the paper demonstrates "the ability to capture the modes, even in the absence of the true oracle".

The work of [6] explicitly addresses imperfect oracles: "Diverse candidates capturing the modes of the imperfect oracle improve the likelihood of discovering a candidate that can satisfy all (or many) evaluation criteria"

**Why GFlowNets Succeed Where RL Fails**
Unlike reinforcement learning methods that maximize expected reward and thus over-exploit Oracle predictions, GFlowNets sample proportionally to reward. This fundamental difference is critical: RL agents trained on noisy oracles converge to exploit spurious high-reward modes, while GFlowNets maintain broad exploration that discovers scenarios the Oracle may undervalue. This is precisely why our REINFORCE and SAC baselines, despite using the identical Oracle, achieve lower median performance and collapse to narrow solution sets (Figure 2, Table 3).

We updated our submission (Section F in the appendix) to better explain these important points.

[1] Hou, Kewei, Lin Peng, and Wei Xiong. Is R-squared a measure of market inefficiency?. 2013.

2] Kelly, Bryan, and Seth Pruitt. Market expectations in the cross‐section of present values. The Journal of Finance, 2013

[3] Bengio, Yoshua, et al. "Gflownet foundations." JMLR, 2023

[4] Jain, Moksh, et al. "Gflownets for ai-driven scientific discovery." Digital Discovery, 2023

[5] Bengio, Emmanuel, et al. "Flow Network based Generative Models for Non-Iterative Diverse Candidate Generation." NeuRIPS, 2021

[6] Jain, Moksh, et al. "Biological sequence design with gflownets." ICML, 2022.

---

### Author Response · Authors · 2025-12-01
**Summary and key information for the Area Chair review**

Dear Area Chair,

Given the unfortunate circumstances surrounding the OpenReview security breach and the measures taken by ICLR, we would like to provide you with key information to assist in your decision-making process.

***Comprehensive Resolution of Core Reviewer Concerns***

Despite the diversity in initial scores, *all reviewers converged on three fundamental concerns*, which we thoroughly addressed in our rebuttals:

**1. Oracle Quality and R² Performance:** Multiple reviewers questioned our Oracle's negative R² values. We demonstrated that: (a) low R² values are standard in financial forecasting literature, with highly influential papers (Hou et al. 2013; Kelly & Pruitt 2013) establishing that R² is independent of model utility in rational financial models; (b) GFlowNets were explicitly designed by Bengio et al. to operate with imperfect oracles—this is a core design principle, not a limitation; and (c) our approach outperformed baselines using the identical oracle, proving our contribution stems from superior exploration capabilities, not oracle quality.

**2. Effectiveness with Imperfect Oracles:** We cited multiple foundational papers by the creators of GFlowNets (Bengio et al., JMLR 2023; Jain et al., ICML 2022) that explicitly state GFlowNets are designed for scenarios with "cheap but inaccurate measurements." Our proportional sampling mechanism discovers multiple plausible modes even with noisy proxies, while RL baselines collapse to narrow solutions—validating our architectural choice.

**3. Real-World Scenario Detection:** We conducted trajectory similarity analysis comparing all methods' top-performing scenarios against actual historical crisis trajectories (2002, 2008, 2021) using Pearson correlation. Our approach achieved the highest similarity in at least one evaluation setting for each crisis, while maintaining superior diversity (higher standard deviation), demonstrating our method discovers a broader array of realistic risks.

***Score Revision Impact***

Reviewer iZtM initially provided a score of 6, then after our detailed responses raised their score to 8, stating: "Thanks for the detailed reply and the released codes. I decided to raise the score." While this score increase was reverted due to the security incident, the reviewer's comment remains on record and reflects their assessment after full consideration of our rebuttals.

Unfortunately, the other reviewers did not respond before the security breach was detected.

***Understanding the Lower Scores***

The two reviewers who assigned lower scores (627W and jQia, both rating 2) raised concerns primarily about oracle quality and questioned whether our approach truly discovers high-risk scenarios. Our rebuttals comprehensively addressed these concerns through: (a) theoretical justification from GFlowNet literature; (b) empirical validation showing superior real-world crisis prediction; and (c) additional financial risk metrics (CVaR, VaR, drawdown statistics) demonstrating our method's practical utility. Notably, Reviewer 627W's confidence level was 2 ("quite likely that you did not understand the central parts"), and Reviewer jQia acknowledged our approach addresses "a highly significant and challenging problem" with "novel problem formulation".

The core disagreement was not about methodological soundness but about understanding the intentional design choice to balance diversity and extreme-value discovery -- a critical requirement in financial risk management where discovering multiple failure modes is more valuable than finding a single worst-case scenario.

***The Case for Careful Consideration***

The diversity of scores reflects the interdisciplinary nature of this work and suggests the paper warrants careful analysis rather than rejection based on numerical averages. We have:

•	First application of GFlowNets to large-scale continuous state spaces in a high-impact domain (financial risk modeling)

•	Comprehensive experimental validation on three major financial crises with 30 independent runs per method

•	Superior performance on both diversity and quality metrics compared to established baselines

•	Demonstrated real-world applicability through crisis trajectory prediction analysis

•	Addressed all technical concerns raised by reviewers with theoretical and empirical evidence

This work represents a significant methodological contribution to both the GFlowNet literature and practical financial risk management, as acknowledged by multiple reviewers including those who gave positive scores.

We respectfully request that you consider the substance of the reviewer discussions, the comprehensive nature of our rebuttals, and the evidence of score improvement (now reverted) when making your decision. We believe the paper makes a strong contribution to the conference and warrants acceptance.

Thank you for your time and consideration during these challenging circumstances.

Sincerely,

The authors

---

### Meta-Review · Area_Chair_pQ6h · 2026-01-03

**Summary:**

This paper presents an interesting application of GFlowNets to a challenging and important problem. However, the review process revealed severe, consensus concerns regarding the validity of the experimental foundation primarily the poor-performing Oracle and the unconventional evaluation setup. These concerns strike at the heart of the paper's claims. The authors' rebuttal, while diligent, failed to adequately resolve these critical issues. Consequently, the evidence presented is insufficient to support acceptance. The paper requires a more rigorous validation framework and a more convincing demonstration of its core contributions before it would be suitable for publication at ICLR.

**Reviewer Concerns:**

The authors' rebuttal was comprehensive but largely defensive, reiterating justifications rather than providing transformative new evidence.

- Addressed Concerns: The rebuttal effectively clarified that GFlowNets are designed for imperfect oracles and provided additional financial risk metrics (CVaR, VaR). It also corrected some presentation errors.
- Outstanding & Fatal Concerns: The fundamental issues were not resolved:
    1. Oracle Validity: The defense that negative R² is "common" in finance does not alleviate the concern that the reward function may be uninformative. The success of the method remains contingent on a signal whose utility is seriously in doubt.
    2. Realism & Setup: The trajectory correlation analysis is a step, but does not substantively address the core criticism about the Oracle's validity or the unconventional train/test mismatch. The setup remains a significant barrier to evaluating the paper’s claims.

**Reviewer Scores:**

*   Reviewer iZtM (Initial: 6): Likely would have maintained or slightly increased their score (to 8). They were initially positive, raised their score post-rebuttal (before reversion), and their primary concern (code availability) was met. However, the unresolved core issues might have tempered enthusiasm.
*   Reviewer PPUF (Initial: 6): Likely would have lowered their score (to 5 or 4). Their detailed questions about realism, reward consistency, and evaluation breadth were only partially addressed. The persistence of the fundamental oracle issue would likely reduce their confidence in the findings.
*   Reviewer 627W (Initial: 2) & jQia (Initial: 2): Both would have maintained their reject scores. Their critiques targeted the foundational soundness of the experimental design and the interpretation of results. The rebuttal did not provide the strong new validation (e.g., compelling evidence of realism, justification for the oracle mismatch, clear superiority on tail metrics) needed to overturn their assessment.

---

### Decision · Program_Chairs · 2026-01-26

Reject